# Protein allocation and utilization in the versatile chemolithoautotroph *Cupriavidus necator*

**Michael Jahn\*, Nick Crang, Markus Janasch, Andreas Hober, Björn Forsström, Kyle Kimler[†], Alexander Mattausch[‡], Qi Chen, Johannes Asplund-Samuelsson, Elton Paul Hudson\***

School of Engineering Sciences in Chemistry, Biotechnology and Health, Science for Life Laboratory, KTH – Royal Institute of Technology, Stockholm, Sweden

**\*For correspondence:**
michael.jahn@scilifelab.se (MJ);
paul.hudson@scilifelab.se
(EPaulH)

**Present address:** [†]The Broad Institute and Boston Children's Hospital, Boston, United States; [‡]Institute of Pharmacy and Molecular Biotechnology, Heidelberg University, Heidelberg, Germany

**Competing interest:** The authors declare that no competing interests exist.

**Abstract** Bacteria must balance the different needs for substrate assimilation, growth functions, and resilience in order to thrive in their environment. Of all cellular macromolecules, the bacterial proteome is by far the most important resource and its size is limited. Here, we investigated how the highly versatile 'knallgas' bacterium *Cupriavidus necator* reallocates protein resources when grown on different limiting substrates and with different growth rates. We determined protein quantity by mass spectrometry and estimated enzyme utilization by resource balance analysis modeling. We found that *C. necator* invests a large fraction of its proteome in functions that are hardly utilized. Of the enzymes that are utilized, many are present in excess abundance. One prominent example is the strong expression of CBB cycle genes such as Rubisco during growth on fructose. Modeling and mutant competition experiments suggest that $CO_2$-reassimilation through Rubisco does not provide a fitness benefit for heterotrophic growth, but is rather an investment in readiness for autotrophy.

## Editor's evaluation

This work combines elegant experimental approaches with modelling predictions to study metabolic adaptations in the bacterium *Cupriavidus necator*, a microorganism of interest given its metabolic versatility and potential industrial applications. This manuscript will be interesting for microbiologists and systems biologists who want to understand how protein production and economy and enzyme utilization differs in a versatile microorganism in different conditions

## Introduction

*Cupriavidus necator* (formerly *Ralstonia eutropha*) is a model aerobic lithoautotroph and formato-troph, and is notable for production of the storage polymer polyhydroxybutyrate (PHB) (*Yishai et al., 2016*; *Brigham, 2019*). *Cupriavidus necator* H16 (hereafter abbreviated *C. necator*) is a soil-dwelling bacterium with a large genome (~6600 genes) distributed on two chromosomes and one megaplasmid (*Pohlmann et al., 2006*). It features a wide arsenal of metabolic pathways for xenobi-otics degradation, hydrogen and formate oxidation, carbon fixation via the Calvin-Benson-Bassham (CBB) cycle, and utilization of nitrate/nitrite as alternative electron acceptors (de-nitrification; *Cramm, 2009*). Several operons for substrate assimilation are present in multiple copies, often on different chromosomes (e.g. *cbb* operon, hydrogenases, formate dehydrogenases). A detailed reconstruction of its metabolic network suggested that it can metabolize 229 compounds (*Park et al., 2011*). Interestingly, *C. necator* prefers organic acids as growth substrate over sugars. The only sugars that support growth are fructose and N-acetylglucosamine (*Cramm, 2009*), which are metabolized via

the Entner-Doudoroff (ED) pathway (*Alagesan et al., 2017*). Although the metabolic versatility of *C. necator* is interesting from a biotechnological point of view, this benefit could come at a considerable cost for the cell. For example, it is not known if the expression of the various substrate assimilation pathways is efficiently regulated under different conditions, and if gene expression is optimal to maximize growth or rather another trait such as environmental readiness. The 'cellular economy' concept entails that an organism has a limited pool of (enzyme) resources and must re-allocate resources to different functions in order to meet the current environmental needs (*Molenaar et al., 2009*; *Scott et al., 2014*; *Hui et al., 2015*). A prime example is the switch from energy-efficient, high-enzyme-cost respiration to energy-inefficient, but low-enzyme-cost fermentation during overflow metabolism (*Basan et al., 2015*, *Sánchez et al., 2017*). The protein economy has been studied experimentally and with dedicated metabolic models in heterotrophic microorganisms like *E. coli* (*Scott et al., 2014*; *O'Brien et al., 2016*) and *S. cerevisiae* (*Metzl-Raz et al., 2017*; *Sánchez et al., 2017*). More recently, resource allocation was studied in photoautotrophic bacteria (*Synechocystis* sp.) (*Jahn et al., 2018*; *Zavřel et al., 2019*). There, a large investment in the $CO_2$-fixation (2–7% protein mass is Rubisco) and photosynthesis machinery (20–40% protein mass are antennae and photosystems) may reduce proteome space for ribosomes, resulting in lower growth rates than heterotrophs.

Previous studies of *C. necator* grown in different trophic conditions have shown that gene expression is regulated in a condition-dependent manner (*Schwartz et al., 2009*; *Kohlmann et al., 2011*; *Kohlmann et al., 2014*). For example, CBB cycle genes are strongly expressed during autotrophic growth but were also upregulated on fructose (*Shimizu et al., 2015*), prompting the question of whether such expression confers any evolutionary advantage. To date, protein allocation and utilization has not been investigated. It is unclear if and how *C. necator* would reallocate protein resources when confronted with different types or degrees of substrate limitation, or to what extent a versatile soil bacterium would express unutilized or underutilized proteins. To address these questions, we designed a multivariate set of growth experiments. *C. necator* was cultivated in bioreactors at steady state conditions using four limiting substrates and five different growth rates. We quantified the cellular proteome using LC-MS/MS and trained a genome-scale resource allocation model with our data (*Bulović et al., 2019*, *Goelzer et al., 2015*). We found that *C. necator* allocates its resources in response to the imposed environmental challenges, but invests more than 40 % of its protein mass in genes that are either unlikely to be utilized or have no known function. Enzyme utilization in the central carbon metabolism was markedly different between pathways, with enzymes in the proximity of substrate assimilation (upper glycolysis, CBB cycle) showing higher variability, higher absolute abundance, and higher utilization than enzymes involved in supply of biomass precursors (tricarboxylic acid [TCA] cycle, pyruvate metabolism). $CO_2$-assimilation enzymes expressed in heterotrophic growth regimes were unlikely to provide a fitness benefit.

## Results

### *C. necator* expresses most of its annotated genes

In order to access cellular states that were optimally acclimated to a nutrient limitation, we cultivated *C. necator* in chemostat bioreactors. We selected four limiting growth substrates as interesting entry points to metabolism (*Figure 1A*). Fructose was chosen as a glycolytic substrate because *C. necator* does not naturally utilize glucose (*Orita et al., 2012*). It is taken up via a specific ABC transporter and metabolized in the ED pathway. Succinate was chosen as an entry point to the TCA cycle. Formate was chosen because formatotrophic growth closely resembles lithoautotrophic growth regarding the utilized enzymes (*Cramm, 2009*). Formate (COOH⁻) is first oxidized by formate dehydrogenases (FDH) to $CO_2$ with simultaneous reduction of $NAD^+$ to NADH. The $CO_2$ is then fixed via the CBB cycle. Finally, growth on fructose with limiting ammonium was chosen as we expected a dedicated response to N-limitation by adjustment of gene expression and flux ratios between different pathways. For each limitation, four independent bioreactor cultivations were performed with dilution rate (equalling growth rate μ) increasing step-wise from 0.05 to 0.1, 0.15, 0.2, and 0.25 $hr^{-1}$ (*Figure 1—figure supplement 1*) and subsequent sampling for proteomics. The substrate limitation in chemostats was verified by determining the residual carbon concentration in culture supernatants using HPLC (*Figure 1—figure supplement 1*). For ammonium limitation, a high concentration of residual fructose was determined, as expected when nitrogen is limiting. All other conditions showed no or very low

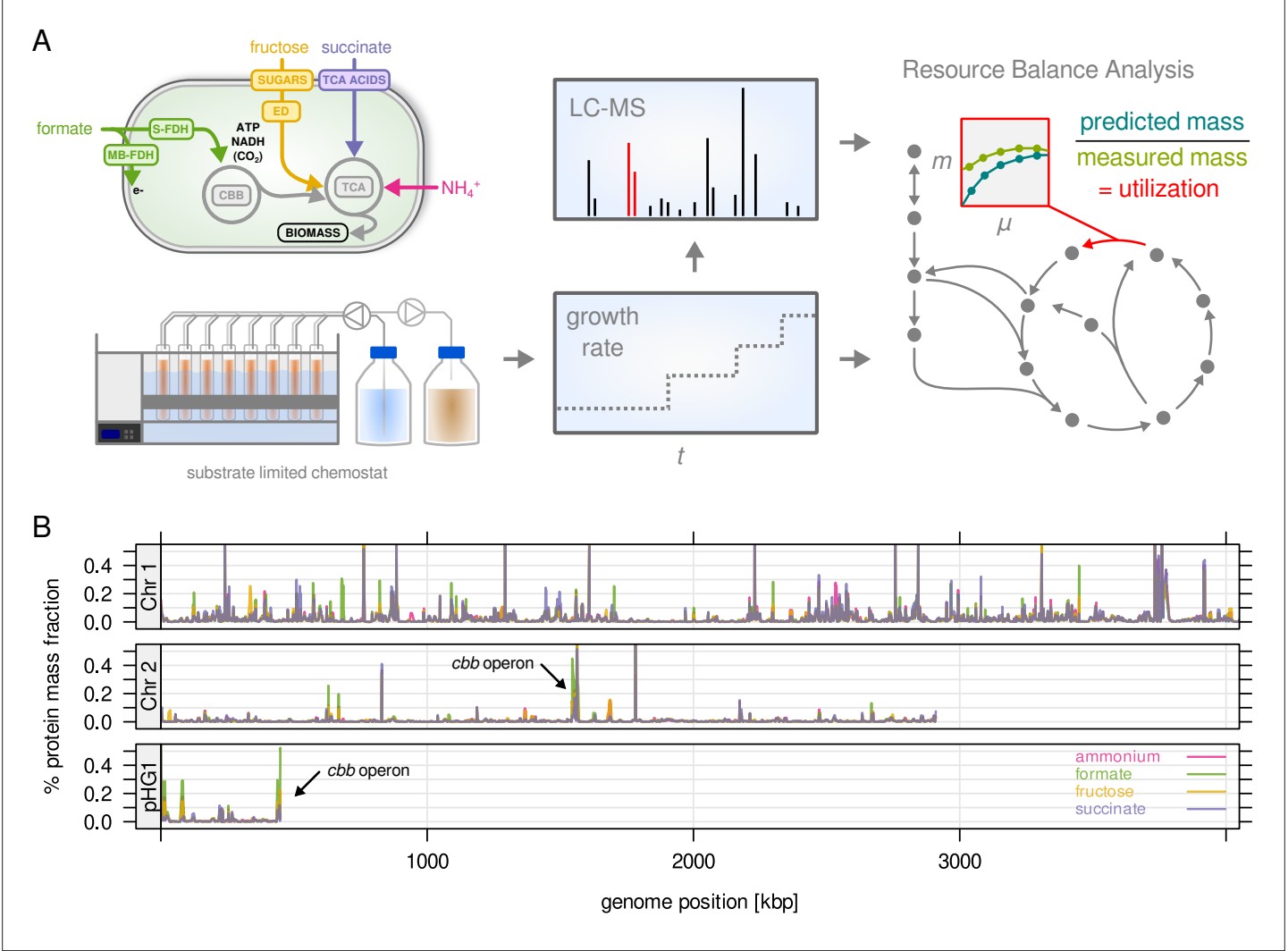

**Figure 1.** *C. necator* expresses most of its annotated genes. (**A**) Four different limitations were chosen covering different entry points to central metabolism. Cells were cultivated in chemostat bioreactors and dilution rate (equals growth rate) was stepwise increased from 0.05 to 0.25 hr$^{-1}$. The proteome was analyzed by LC-MS/MS. Enzyme abundance was used to constrain a resource balance analysis (RBA) model, and enzyme utilization was investigated for the different limitations. (**B**) Protein mass fraction (%) of all proteins (5357) mapped to their respective genes on chromosome 1, 2, and megaplasmid pHG1 (mean of four substrate limitations, μ = 0.25 hr$^{-1}$). Density is mean protein mass fraction for a sliding window of five genes. The genes of the cbb operon (arrows) are the most expressed regions on chromosome 2 and pHG1.

The online version of this article includes the following figure supplement(s) for figure 1:

**Figure supplement 1.** Cultivation parameters for substrate-limited chemostats.

**Figure supplement 2.** Biomass and PHB content of *C. necator* for different limiting conditions.

**Figure supplement 3.** Overview about mass spectrometry based protein quantification.

concentration of residual substrate. Quantification of dry cell weight (DCW) and PHB content revealed that only ammonium-limited cells produced a significant amount of PHB, approximately 80 % of total biomass for the strongest limitation (μ = 0.05 hr$^{-1}$, *Figure 1—figure supplement 2*).

We analyzed the proteome of *C. necator* for all conditions of the chemostat cultivations (four substrate limitations, five growth rates, four biological replicates). We employed a label-free quantification strategy with a feature propagation approach, allowing us to significantly increase the coverage of protein quantification (*Weisser and Choudhary, 2017*). More than 4000 proteins were quantified in each individual sample (*Figure 1—figure supplement 3A*). Altogether, 5357 proteins out of 6614 annotated genes were quantified in at least one condition (81.0 %), and 4260 proteins were quantified with at least two peptides (*Figure 1—figure supplement 3B*). The proteomics data can be

accessed through an interactive web application at https://m-jahn.shinyapps.io/ShinyProt. Based on the distribution of protein abundance 99 % of the proteome by mass was quantified. An analysis of sample similarity based on expression revealed that low growth rates are more similar to each other, and that growth on formate is most unlike the other conditions (*Figure 1—figure supplement 3C*). Gene expression in terms of proteome mass fraction was unequally distributed over the genome (*Figure 1B*): 78.7 % of protein mass was encoded by chromosome 1, 16.4 % encoded by chromosome 2, and 5.4 % by pHG1. Chromosome 2 and pHG1 thus encode predominantly specialized functions, as predicted by *in silico* analyses (*Pohlmann et al., 2006*; *Fricke et al., 2009*). On chromosome 2, highly expressed genes were the *cbb* operon (CBB Cycle, pentose phosphate pathway (PPP), *Figure 1—figure supplement 3D*), glycolysis-related genes (*pgi, zwf*), and the methionine synthase *metE*. On pHG1, highly expressed were the second copy of the *cbb* operon as well as *hox/hyp* operons (soluble and membrane bound hydrogenases, up to 3 % of proteome by mass). The majority of pHG1 encoded protein mass is therefore related to autotrophic growth. Note that the two copies of the *cbb* operon are 99 % identical on amino acid sequence level and can not be distinguished well by LC-MS/MS (abundance of ambiguous peptides was allocated to both copies). Promoter activity studies have shown that expression levels from both operons were similar (*Gruber et al., 2017*). As we also culti- vated *C. necator* on formate, we were interested in the expression of formate dehydrogenase (FDH) genes (*Figure 1—figure supplement 3E*). *C. necator* is equipped with two types of FDH, soluble S-FDH (operons *fds* and *fdw* on chromosome 1 and 2, respectively) and membrane-bound M-FDH (*fdo* and *fdh* operons, the latter present in two copies on chromosome 1 and 2, respectively). In contrast to *cbb* genes, which were expressed under both fructose and formate growth, expression of FDHs was induced only during growth on formate, and the soluble dehydrogenase (*fds*) was the predominant form.

## A large fraction of the *C. necator* proteome is not utilized and not essential

We next explored how the proteins of *C. necator* are utilized during the different growth modes. We created a resource balance analysis (RBA) model (*Bulović et al., 2019*) based on a previous genome- scale metabolic reconstruction of *C. necator* (1360 reactions; *Park et al., 2011*). The RBA model predicts optimal flux distributions as in flux balance analysis (FBA), but also takes kinetic parameters and enzyme abundance into account (Materials and methods). DNA replication, transcription, transla- tion, and protein folding were included as lumped reactions (macromolecular machines) with protein subunit composition and rate estimates taken from the literature (Materials and methods, *Supplemen- tary file 1*). Each enzyme or macromolecular machine imparts a protein cost, with the total protein pool being limited. RBA models can predict trade-offs between high- and low-enzyme-cost pathways, increase of ribosome abundance with growth rate, and upper boundaries on growth in substrate- replete conditions (*Goelzer et al., 2015*; *Sánchez et al., 2017*; *Salvy and Hatzimanikatis, 2020*). The *C. necator* RBA model was constrained using a set of parameters obtained from proteomics data, the UniProt database, and literature (Materials and methods, *Figure 2—figure supplement 1*, *Supple- mentary file 1*). A critical parameter for RBA is the enzyme efficiency $k_{app}$ of each reaction, which links the reaction rate to the abundance of its catalyzing enzyme. These were obtained by estimating the metabolic flux boundaries per reaction (using flux sampling), and then dividing maximal flux by unit enzyme allocated to the reaction (*Goelzer et al., 2015*; *Davidi and Milo, 2017*; *Bulović et al., 2019*).

We used the constrained resource allocation model to analyze the non-utilized and the under- utilized fraction of the *C. necator* proteome. The non-utilized proteome fraction consists of enzymes that do not carry flux in any of the tested conditions. To quantify this fraction, we performed a series of RBA model simulations corresponding to the experimental conditions of the chemostats. The model predicted optimal flux distribution and enzyme abundance to maximize growth rate for each of the four different substrate limitations. The model was generally able to reproduce experimentally determined protein allocation using fitted (optimal) $k_{app}$ values (*Figure 2—figure supplement 2A*). However, these simulations may predict one out of many possible solutions to the protein allocation problem. In order to estimate the total number of usable reactions independent from the optimal set of $k_{app}$, we performed 200 simulations per substrate limitation where $k_{app}$ was randomly sampled from the $k_{app}$ distribution. This converged to maximally 550 utilized reactions per condition. (*Figure 2— figure supplement 2B*). In total, 587 of 1360 reactions were utilized at least once in all simulations,

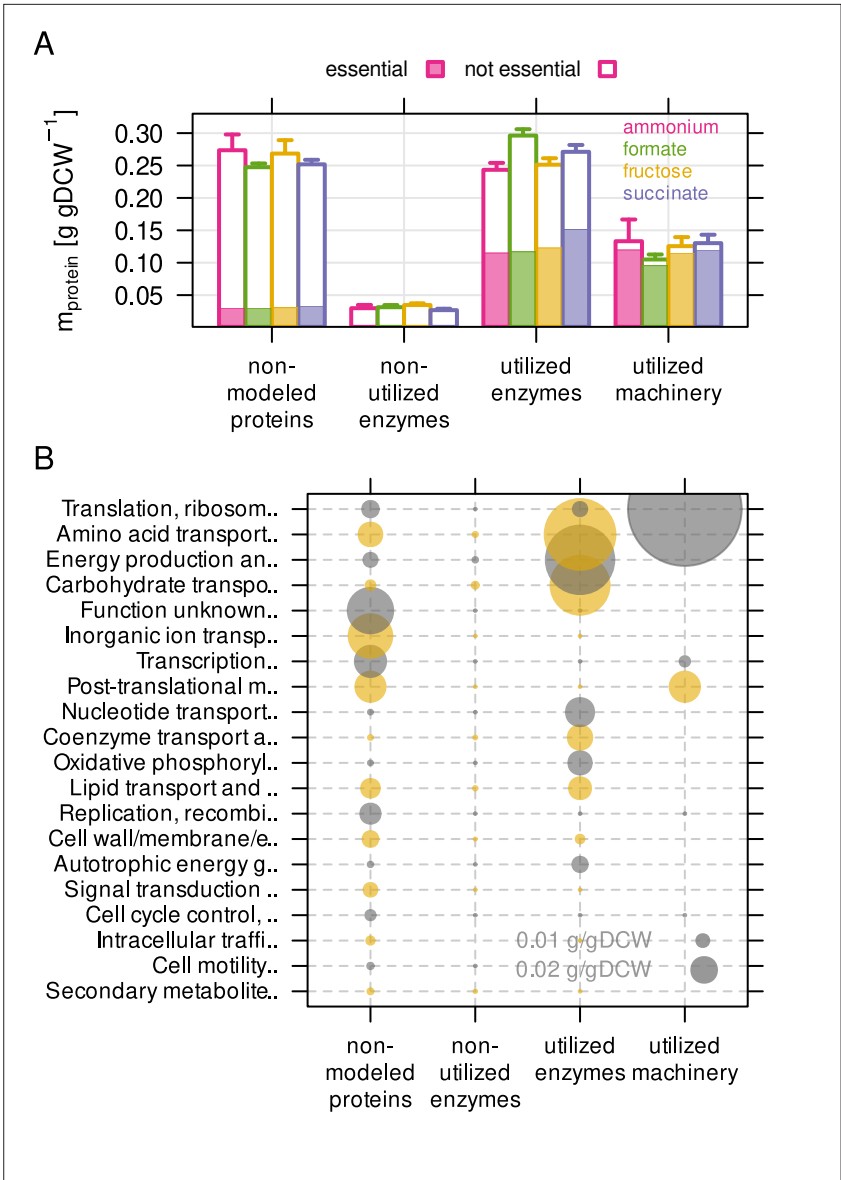

**Figure 2.** The non-modeled and non-utilized proteome of *C. necator* is related to environmental readiness. (**A**) A series of model simulations was conducted with randomly sampled enzyme efficiency $k_{app}$ (n = 200) to obtain the maximum number of potentially utilized reactions in each growth condition. The *C. necator* proteome (5357 proteins) was allocated to each of four utilization categories and protein mass summed up per category. Protein mass encoded by essential genes is indicated as shaded area in bars. Bars represent mean of four biological replicates, whiskers represent standard deviation. (**B**) Average protein mass by utilization category and functional group. Alternating color (gray and yellow) for bubbles are used in alternating rows.

The online version of this article includes the following figure supplement(s) for figure 2:

**Figure supplement 1.** Parameter estimation for RBA model.

**Figure supplement 2.** The unutilized proteome of *Cupriavidus* is related to environmental readiness.

280 reactions were used in all simulations on all substrates (core reactions), and 28 reactions were used in only one particular limitation. We mapped the *C. necator* proteome quantification data onto RBA model reactions to categorize proteins as: (1) not included in the model, (2) included but non-utilized enzymes, (3) utilized enzymes, and (4) utilized machinery (*Figure 2A*). The non-modeled proteome fraction comprised on average 38 % of the proteome mass (0.26 g/gDCW, 4041 proteins), and was slightly dependent on condition. Non-utilized enzymes were low-abundant in mass (0.03 g/

gDCW, 400 proteins) compared to the utilized enzyme fraction (0.27 g/gDCW, 823 proteins). Macromolecular machinery averaged 0.12 g/gDCW for 93 annotated proteins. Non-utilized enzymes were not enriched in a particular functional category, while the non-modeled protein fraction was enriched in functions for transport, transcription (factors), and post-translational modification (*Figure 2B*). A large group of proteins has no annotated function. Taking non-modeled and non-utilized proteins together, 43 % of the *C. necator* proteome (by mass) is unlikely to be utilized in the tested conditions, or involved in processes not covered by the RBA model. We also estimated the protein mass encoded by essential genes per utilization category (*Figure 2A*, shaded area). Gene essentiality was determined by sequencing a randomly barcoded transposon library with 60,000 mutants after growth on rich medium (RB-TnSeq workflow) (*Rubin et al., 2015*; *Wetmore et al., 2015*). Transposon insertion density of a gene was used to sort it into one of three different categories, 'essential' (496 genes), 'probably essential' (149), or 'non-essential' (4,712). On average, 47 % of utilized enzymes (by mass) were encoded by essential genes, while only 19 % and 3 % of the non-modeled and non-utilized protein mass, respectively, was essential. Based on the calculated large fraction of non-modeled and non-utilized proteome, and the observation that approximately half of the enzyme mass is non-essential, we conclude that a large portion of the *C. necator* proteome is associated with nutrient scavenging and regulatory adaptation to new environments.

## Highly utilized enzymes are more abundant, less variable, and often essential

The *under*-utilized proteome fraction is a subset of the utilized fraction. Generally, metabolic flux through a reaction can be correlated to the associated enzyme abundance. The rate of a reaction $v_R$ is then the product of the enzyme efficiency $k_{app}$ and the concentration of the enzyme that catalyzes the reaction ($v_R = k_{app} \cdot [E]$) (*Davidi and Milo, 2017*). Under steady-state conditions, optimal gene expression would adjust enzyme abundance proportional to the flux that it is supposed to carry (metabolic demand), keeping utilization of the enzyme constant. If enzyme abundance and flux do not change proportionally between different conditions or growth rates, utilization changes. To estimate the degree of utilization, we compared experimental protein allocation to model predictions at different growth rates. The RBA model predicts the minimal required enzyme abundance to drive a metabolic reaction, assuming full substrate saturation of the enzyme. Although full saturation of all enzymes is not realistic (*Reznik et al., 2017*; *Janasch et al., 2018*), it is a useful assumption to determine enzyme utilization. Utilization $U_E$ is calculated by dividing the predicted minimal enzyme abundance by the experimentally determined enzyme abundance (*Davidi and Milo, 2017*):

$$U_E \ [\%] \ = \ [E]_{minimal} \ / \ [E]_{measured} \cdot 100$$

We first looked at utilization of the macromolecular machines (*Figure 3—figure supplement 1*). Only two of these, ribosomes and chaperones, had a considerable protein mass allocated to them. The abundance of ribosomal proteins increased linearly with growth rate, as observed in other bacteria (*Scott et al., 2014*; *Peebo et al., 2015*; *Jahn et al., 2018*). The RBA model simulations accurately predicted expansion of ribosomes with increasing growth rate, but failed to predict incomplete reduction of ribosomes at low growth rate (*Figure 3—figure supplement 1B*). This can be explained by the evolutionary benefit that cells gain from keeping a ribosome reserve for nutrient upshifts (*Mori et al., 2017*). The ribosome reserve led to a decrease in utilization at low growth rate regardless of the limiting substrate (*Figure 3—figure supplement 1C*).

Next, we examined metabolic enzyme utilization by comparing experimental and simulated protein abundance. All metabolic reactions/enzymes of the RBA model that had associated proteins quantified by MS were included in the analysis (n = 1012). For each enzyme, the average utilization in the four limiting conditions ($\mu$ = 0.25 hr$^{-1}$) was determined, and then used to group enzymes into three categories: low ($\leq$ 33 %, n = 710), moderate (33–66 %, n = 153) and high utilization ( > 66 %, n = 149). Highly utilized enzymes are therefore predominantly enzymes utilized in several of the four limiting conditions. There were significant differences between these three groups: Highly utilized enzymes were on average more abundant in terms of protein mass (g/gDCW) (*Figure 3A*). We also calculated variability in enzyme abundance by determining the coefficient of variation (CV) of allocated protein mass across the four different conditions (*Figure 3B*). For example, formate dehydrogenase (FDH) was strongly expressed in only one out of four conditions (growth on formate) and therefore showed high

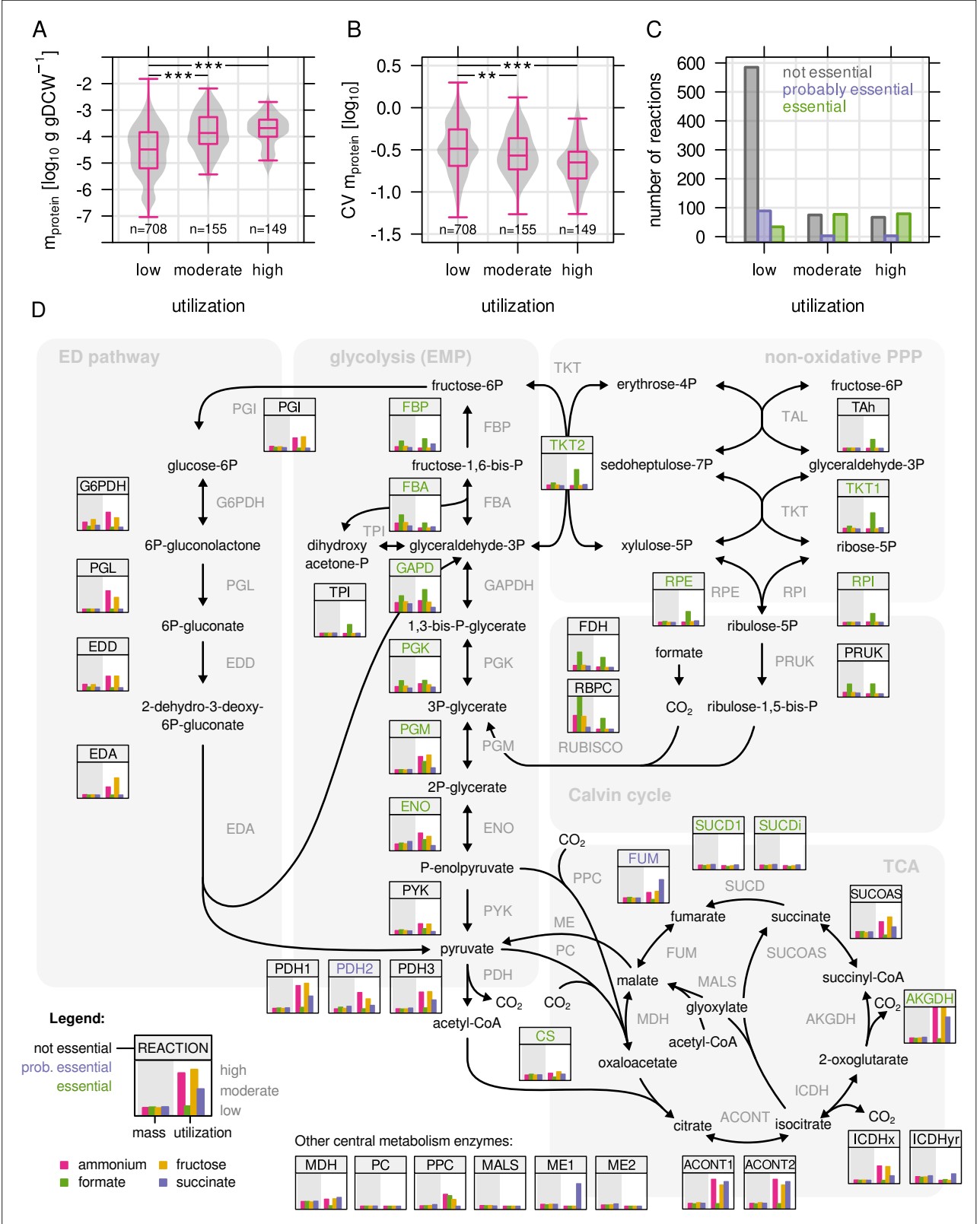

**Figure 3.** Highly utilized enzymes are more abundant, less variable, and often essential. (**A**) Protein mass in g/gDCW allocated to enzymes with low, moderate, and high utilization. Enzymes with moderate and high utilization were significantly more abundant than enzymes with low utilization ($P = 2.2 \times 10^{-16}$ and $3.2 \times 10^{-21}$, respectively; Mann-Whitney U-test, two-sided). (**B**) Coefficient of variation (CV) as a measure of variability in enzyme abundance. Enzymes with moderate and high utilization had significantly lower variability than enzymes with low utilization ($P = 2.1 \times 10^{-3}$ and $1.8 \times 10^{-12}$,

*Figure 3 continued on next page*

*Figure 3 continued*

respectively. Mann-Whitney U-test, two-sided). (**C**) Number of reactions associated with at least one essential gene, or at least one probably essential gene, or no essential gene at all, broken down by utilization. (**D**) Map of *C. necator*'s central carbon metabolism. Inset figures show enzyme abundance and utilization for the four limiting conditions ($\mu = 0.25$ hr$^{-1}$, four biological replicates). Values were rescaled from the respective minimum and maximum to a range of 0–1. Enzyme abbreviations are colored according to essentiality as described in (**C**).

The online version of this article includes the following figure supplement(s) for figure 3:

**Figure supplement 1.** Utilization of molecular machinery related to central dogma depends on growth rate.

**Figure supplement 2.** Enzyme mass and utilization of the PHB biosynthesis pathway.

variability (CV = 1.25), and low average utilization (23 %). Altogether, variability was significantly lower for moderately and highly utilized enzymes. These observations support the notion that *C. necator* optimizes the cost-benefit ratio of gene expression by keeping utilization high for highly abundant enzymes. Similarly, low variation in gene expression of highly utilized enzymes could provide a fitness benefit in conditions changing on a short time scale. Constitutive expression of such genes can buffer substrate and metabolite surges. Finally, we wondered if utilization of enzymes is also correlated to essentiality of the associated gene(s) as determined by RB-TnSeq from our transposon mutant library. Enzymes were sorted into, 'essential', 'probably essential', or 'non-essential' based on the essentiality of their associated genes (Materials and methods, *Figure 3C*). We found that enzymes with intermediate and high utilization were more likely to be encoded by an essential gene compared to lowly utilized enzymes.

A closer inspection of the central carbon metabolism of *C. necator* revealed that enzyme abundance and utilization was markedly different between major pathways (*Figure 3D*). The enzymes in upper glycolysis (PGK, GAPDH, FBA, FBP) and the CBB cycle showed a clear condition-dependent trend, with high expression and utilization on formate, and low expression and utilization on succinate. The enzymes of lower glycolysis (PGM, ENO, PYK, PDH) showed low expression, low variability and moderate to high utilization in all conditions, clearly distinct from the enzymes in upper glycolysis. This trend continued with reactions down-stream of glycolysis/gluconeogenesis, such as the reactions of pyruvate metabolism and the TCA cycle (low, invariable expression). The ED pathway was only expressed and utilized when fructose was used as carbon source. Gene expression regulation in *C. necator* is thus hierarchically organized: Enzymes close to the entry point of substrates into central metabolism are expressed 'on demand', and show high variability, high absolute abundance, and high utilization in some growth regimes. Enzymes downstream of substrate assimilation show lower expression and variability, perhaps owing to their universal role in providing biomass precursors (TCA, pyruvate metabolism). A lower protein investment per catalytic activity allows for larger reserves of these enzymes. The low utilization of many TCA and pyruvate metabolism enzymes may provide a benefit for robustness by avoiding full saturation. We also inspected the enzymes of the PHB biosynthesis pathway (*Figure 3—figure supplement 2*), Acetyl-CoA acetyltransferase (*phaA*), Acetoacetyl-CoA reductase (*phaB*), and PHB synthase (*phaC*). PhaA and *phaB* were highly abundant while *phaC* abundance was comparatively low. All enzymes showed a similar pattern of increased expression with decreasing growth rate regardless of the limiting substrate. However, only nitrogen limitation triggered significant PHB production which is reflected in the strong utilization of the PHB biosynthesis pathway in this condition.

## Autotrophy-related enzymes are largely underutilized

The high average abundance and variability of the CBB cycle enzymes is particularly interesting. While phosphoribulokinase (PRUK) and Rubisco (RBPC) are specific for the purpose of $CO_2$-fixation, the other enzymes overlap with sugar phosphate metabolism (glycolysis/gluconeogenesis, pentose phosphate pathway) providing precursors that are essential for growth. We wondered if the expression of these enzymes is optimally regulated based on the metabolic demands of the four different substrate limitations. We compared the predicted (optimal) abundance with the experimentally measured abundance for important enzymes of the CBB cycle (*Figure 4A*). On formate, the protein concentration of these enzymes increased with growth rate and therefore estimated flux, correlating with RBA model predictions. A positive correlation was also found for fructose-limited growth, but a negative correlation for succinate and ammonium limitation. Rubisco was highly abundant even during growth on

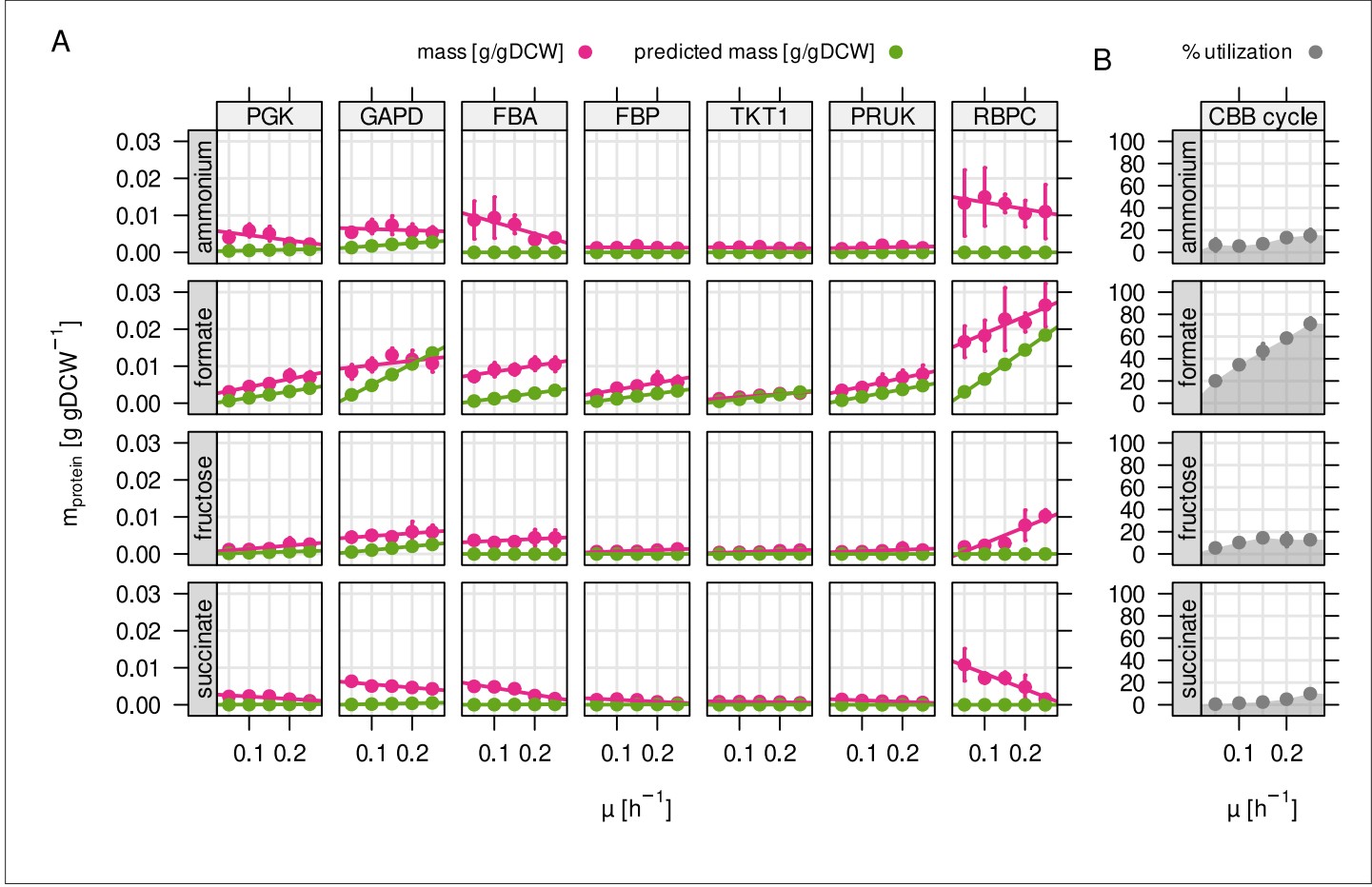

**Figure 4.** Autotrophy-related enzymes are largely underutilized. (**A**) Experimentally determined and model-predicted protein concentration for the seven most abundant enzymes of the CBB cycle (points and error bars represent mean and standard deviation of four biological replicates, respectively). PGK, phosphoglycerate kinase; GAPD, glyceraldehyde-3-phosphate dehydrogenase; FBA, fructose bisphosphate aldolase; FBP, fructose bisphosphatase; TKT1, transketolase; PRUK, phosphoribulokinase; RBPC, ribulose bisphosphate carboxylase. (**B**) Total utilization of the enzymes in (**A**). Utilization was calculated as the sum of predicted (optimal) enzyme abundance divided by the sum of experimentally measured abundance.

The online version of this article includes the following figure supplement(s) for figure 4:

**Figure supplement 1.** Protein mass of the five most abundant glycolytic enzymes broken down by genetic locus.

fructose where the model did not predict flux through the CBB cycle (up to 0.02 g/gDCW or 3 % of the proteome by mass). With the exception of Rubisco and PRUK, the CBB cycle enzymes are encoded by three different copies on the *C. necator* genome. Two of these are arranged in the *cbb* operons on chromosome 2 and pHG1, while the respective third copy on chromosome one is the evolutionarily most ancestral (*Pohlmann et al., 2006*; *Fricke et al., 2009*). Expression of the ancestral enzymes is regulated differently than the *cbb* operons, with lower average protein abundance that is independent of substrate and growth rate (*Figure 4—figure supplement 1*).

When estimating the utilization of *cbb* enzymes, we found that utilization was high for formate due to the obligatory flux through the CBB cycle, but low for other conditions (*Figure 4B*). It was not zero for some reactions that are required to drive lower glycolysis for catabolism of fructose (PGK, GAPDH), or the non-oxidative PPP for the purpose of nucleotide synthesis (transketolase reactions TKT1/2). We conclude that *C. necator* keeps large amounts of underutilized CBB enzymes (0.024–0.04 g/gDCW, or 3.5 % to 5.9 % of the proteome depending on substrate) whose abundance is not warranted by the expected fluxes from glycolysis/gluconeogenesis or nucleotide biosynthesis. The underutilized enzyme mass may be in preparation for autotrophic or formatotrophic growth, even when such substrates are not in reach. The *cbb* operon also encodes several accessory enzymes that were quantified but where utilization could not be estimated (*cbbX, cbbY, cbbZ*, **Figure 1—figure supplement**

3). The most notable example is *cbbZ*, encoding the key enzyme of the 2-phosphoglycolate (2-PGly) salvage pathway (*Claassens et al., 2020*). Phosphoglycolate salvage becomes necessary when the intracellular $CO_2$ concentration is low and the Rubisco oxygenation reaction is more prominent, producing 2-PGly. It is not known if growth on formate leads to considerable flux towards 2-PGly, but the ratio of substrate specificities for $CO_2$ and $O_2$ for *C. necator*'s Rubisco (IC type) of 75 suggests low 2-PGly synthesis compared to 3-PGA (*Horken and Tabita, 1999*). We found that none of the primary 2-PGly salvage enzymes (glycerate pathway) were upregulated on formate, and the knock-out of these enzymes had no effect on growth. This suggests that phosphoglycolate salvage does not play a vital role during growth on formate.

## Reassimilation of $CO_2$ is unlikely to provide a fitness benefit on fructose

*C. necator* appears to keep large amounts of Rubisco (and other CBB cycle enzymes) under-utilized during heterotrophic growth. However, the RBA model finds only optimal flux solutions that maximize growth while other objectives are also possible. It was shown that *C. necator* fixes emitted $CO_2$ via Rubisco during growth on fructose (*Shimizu et al., 2015*). Knock out of Rubisco reduced PHB yield on fructose by 20 % during nitrogen starvation. We wondered if activity of the CBB cycle could improve total carbon yield (biomass including PHB) at the cost of lower growth rate, representing a yield-growth rate trade-off. To test if reassimilation of emitted $CO_2$ improves carbon yield, we performed RBA model simulations on fructose and forced flux through Rubisco (*Figure 5A-F*). We simulated five different $CO_2$ fixation rates (0–5 mmol gDCW$^{-1}$ hr$^{-1}$) at a fructose uptake rate of 4.0 mmol gDCW$^{-1}$ hr$^{-1}$. However, neither biomass yield nor growth rate was improved in any of the simulations (*Figure 5B and C*). Metabolic flux was diverted from the ED pathway towards the non-oxidative PPP in order to provide ribulose-5-phosphate precursors for $CO_2$ fixation (*Figure 5D*). Simultaneously, the high energy requirement for $CO_2$ fixation led to higher flux through the TCA cycle in order to generate additional NADH and ATP. Respiration and $O_2$ consumption was also predicted to increase, while no net reduction of $CO_2$ emission was found. Simulations suggested instead that the cells emit more $CO_2$ when $CO_2$ fixation is enforced, an apparent paradox caused by the lack of additional energy. This can also be inferred from the similar degree of reduction for fructose and biomass (4.0 and 4.12 per C-mol, respectively, *Shuler and Kargi, 2002*), leaving no extra redox power for gratuitous $CO_2$ reassimilation.

We then tested experimentally if expression of CBB genes conveys a fitness benefit during growth on different carbon sources. To this end, the barcoded transposon library (pool of 60,000 mutants) was cultivated in fructose-, succinate-, and formate-limited chemostat bioreactors (dilution rate of 0.1 hr$^{-1}$). The continuous feed fixes the growth rate and selects cells with higher substrate affinity or biomass yield (*Wides and Milo, 2018*). The composition of the mutant pool was checked after 8 and 16 generations of growth using next-generation sequencing. The fitness contribution of each gene was estimated by the degree of enrichment or depletion of mutants over time (*Wetmore et al., 2015*). Surprisingly, fitness of *cbb* mutants was largely unchanged, even during growth on formate where the activity of the CBB cycle is essential for growth (*Figure 5E*). These results show that knockout of *cbb* genes are fully compensated by the second copy of the *cbb* operon. A notable exception was *cbbR*, the transcriptional regulator of the *cbb* operon. Knockout of *cbbR* leads to a 100-fold down-regulation of *cbb* gene expression (*Shimizu et al., 2015*). Although two copies of the *cbbR* regulator are present, only the chromosome 2 copy is functional, the pHG1-encoded copy is inactive due to a 26 bp deletion (*Bowien and Kusian, 2002*). CbbR mutants had a strong fitness penalty on formate (*Figure 5F*, fitness ≤ –6) but no significant fitness penalty on fructose or succinate; the observed fitness effects of –1 to –2 were within the typical variation for neutral genes. This suggests that the activity of the CBB cycle is either neutral to growth or the effect is too small to detect with our method. We reproduced these experiments with a cultivation regime that primarily selects for faster growth rate (medium pulses every 2 hr) and obtained similar results (*Figure 5—figure supplement 1*). We conclude that (re-) fixation of $CO_2$ during hetero-trophic growth is unlikely to convey a growth benefit without additional energy, such as from $H_2$ oxidation or during growth on substrates that are more reduced than biomass. We hypothesize that the up-regulation of Rubisco on fructose is a 'byproduct' of up-regulation of other glycolysis related genes of the *cbb* operon.

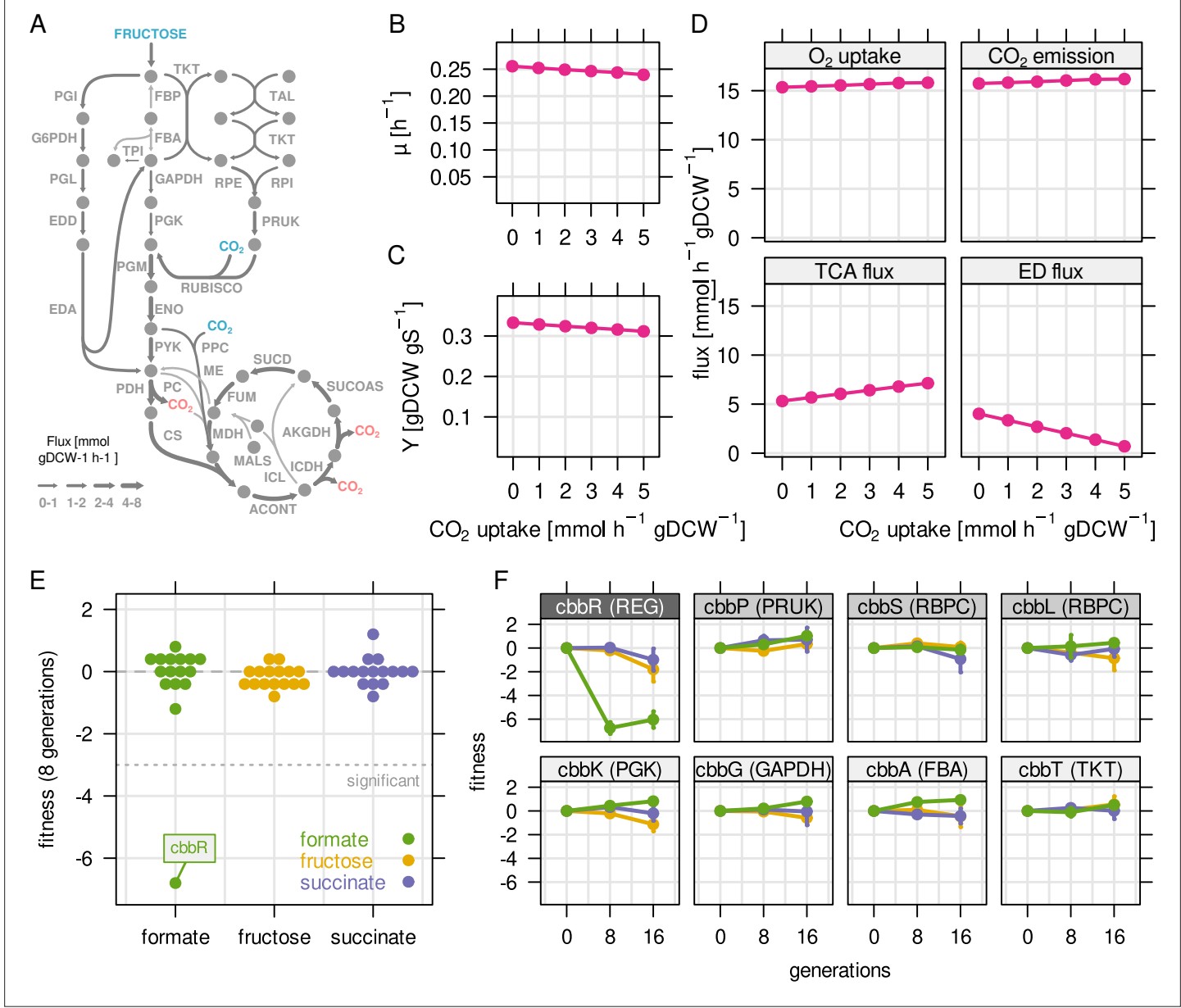

**Figure 5.** Reassimilation of $CO_2$ is unlikely to provide a fitness benefit on fructose. RBA model simulations were performed for a fixed fructose uptake rate combined with five different $CO_2$ fixation rates. (**A**) Example metabolic flux map for a fructose uptake rate of 4.0 mmol gDCW$^{-1}$ hr$^{-1}$ and $CO_2$ fixation rate of 3 mmol gDCW$^{-1}$ hr$^{-1}$. Blue - uptake of fructose and $CO_2$, red - emission of $CO_2$. (**B**) Predicted growth rate μ. (**C**) Biomass yield Y in gDCW g fructose$^{-1}$. (**D**) Net flux through selected reactions for the same simulations as in (**B**) and (**C**). For the TCA cycle, flux through citrate synthase was used as a proxy. For the Entner-Doudoroff (ED) pathway, flux through 6-phosphogluconolactonase (EDD) was used as a proxy. (**E**) Fitness for all *cbb* genes determined by growth competition of a barcoded transposon knockout library on three different substrates. (**F**) Fitness over time for selected *cbb* genes of the pHG1 encoded operon, except *cbbR* which is located on chromosome 2. Chromosome 2 encoded *cbb* genes are not shown due to low transposon insertion frequency. Points and error bars represent mean and standard deviation of four biological replicates, respectively. Grayscale labels indicate role in CBB pathway: dark gray - transcriptional regulator, moderate gray - specific for $CO_2$ fixation, light gray - overlapping role in glycolysis/ CBB cycle.

The online version of this article includes the following figure supplement(s) for figure 5:

**Figure supplement 1.** Fitness of selected genes obtained from cultivation of a barcoded transposon mutant library.

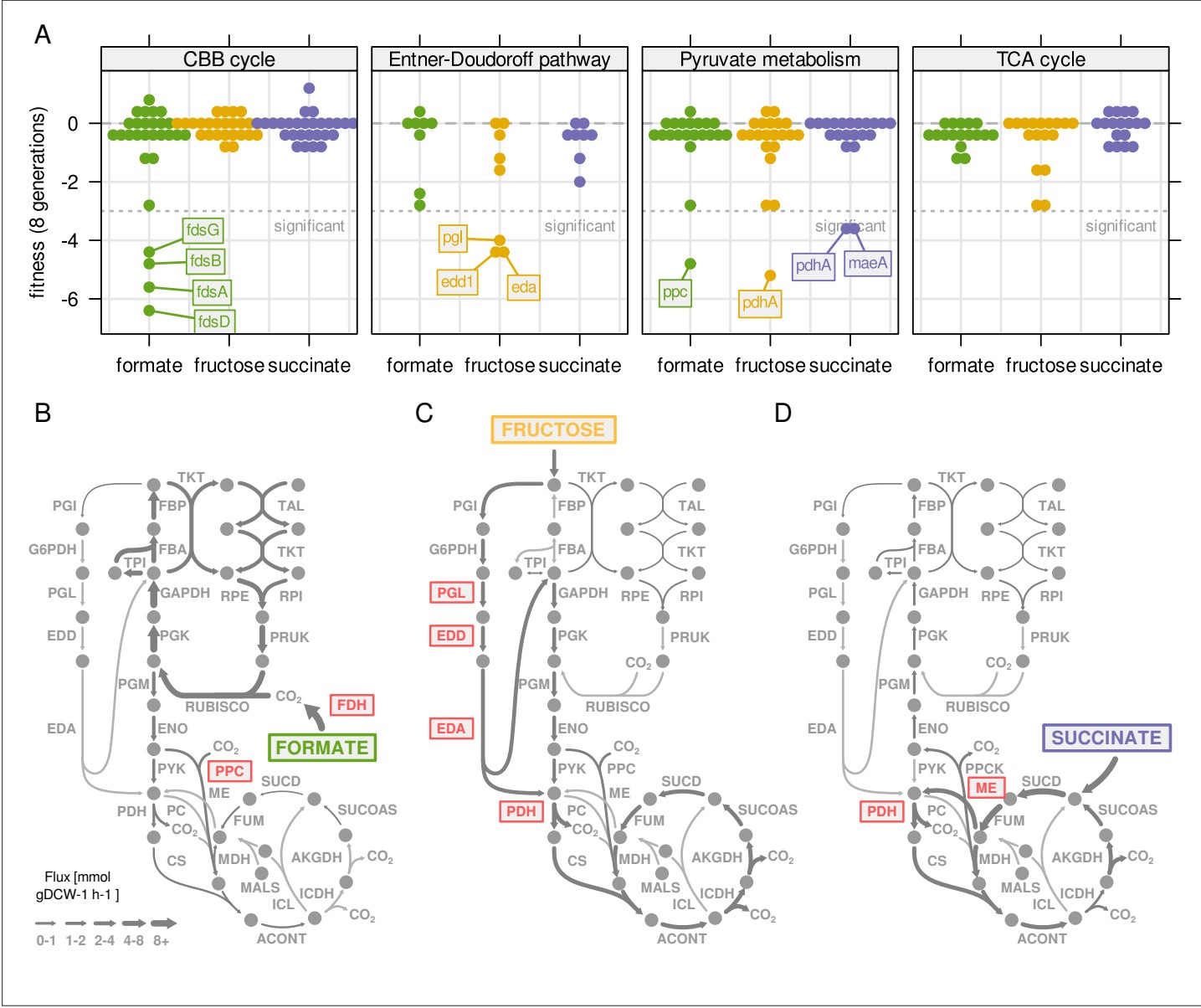

**Figure 6.** Central metabolism enzymes are highly redundant in C. necator. (**A**) Fitness for all central carbon metabolism genes associated with the reactions in *Figure 3D*. Fitness was determined by growth competition of a barcoded transposon knockout library on three different substrates. Genes are broken down by pathway. Dotted line - fitness ≤ –3 was regarded as significant. A summary of all reactions with significantly changed fitness is available in *Supplementary file 2*. (**B**) Metabolic flux map for growth on formate. RBA simulation with formate uptake rate of 62 mmol gDCW$^{-1}$ hr$^{-1}$. Red - reaction where annotated genes show significantly reduced fitness in growth competition from (**A**). (**C**) Same as (**B**) for fructose with uptake rate of 4.0 mmol gDCW$^{-1}$ hr$^{-1}$. (**D**) Same as (**B**) for succinate with uptake rate of 8.3 mmol gDCW$^{-1}$ hr$^{-1}$.

The online version of this article includes the following figure supplement(s) for figure 6:

**Figure supplement 1.** Fitness for all central carbon metabolism genes associated with the reactions in *Figure 3D*.

## The central metabolism of *C. necator* is highly redundant

We have previously established that several enzymes in the central carbon metabolism of *C. necator* are encoded by strictly essential genes (*Figure 3D*). However, most reactions are annotated with more than one (iso-) enzyme. We therefore expanded our gene fitness analysis to all enzymes of central carbon metabolism in order to find conditionally essential genes. The reactions of central carbon metabolism were grouped into four different pathways, CBB cycle including FDH, ED pathway, pyruvate metabolism and TCA cycle, and the fitness of all genes associated with these reactions was quantified (*Figure 6A*, replication experiment in *Figure 6—figure supplement 1*). The majority

of genes showed no significant fitness penalty (or benefit) when knocked out. Only a few genes showed a significant decrease in fitness, and the effect on fitness was substrate-specific. Four genes encoding subunits of a soluble FDH (*fdsABDG*) showed significantly reduced fitness on formate. This demonstrates that *fds* encodes the dominant FDH activity (*Figure 6B*, *Figure 1—figure supplement 2*). No other annotated FDH genes had a similar fitness penalty (*Supplementary file 2*). Another conditionally essential gene on formate was *ppc,* encoding the PEP-carboxylase (PPC). The reaction has no other annotated (iso-) enzymes and was predicted by RBA to carry substantial flux towards the TCA cycle on formate and fructose, but not on succinate (*Figure 6B–D*). The fitness penalty of *ppc* knock-out mutants reflected the relative importance of the reaction for growth on the different substrates (formate: –4.2, fructose: –2.7, succinate: –0.1, *Supplementary file 2*). On fructose, genes for four consecutive enzyme reactions had significantly reduced fitness, *pgl*, *edd1,* and *eda* from the ED pathway, as well as *pdhA* encoding the E1 component of pyruvate dehydrogenase (*Figure 6C*). For EDD another isoenzyme is annotated (*edd2*) that could not compensate for the *edd1* knockout. For PDH, five alternative loci are annotated, all of which did not rescue *pdhA* knockout (*Supplementary file 2*). On succinate, only two gene knockouts have significantly reduced fitness, malic enzyme *maeA* and *pdhA*. Both associated reactions (ME and PDH) carry significant flux on succinate according to RBA simulations (*Figure 6D*). Malic enzyme has one more annotated gene, *maeB*, with different cofactor specificity (NADPH instead of NADH), which could not compensate for the loss of *maeA* (*Supplementary file 2*). We conclude that the central carbon metabolism of *C. necator* has a very high degree of redundancy. Apart from a core set of essential genes encoded on chromosome 1, many enzyme functions can be compensated by alternative copies. The genes that were found to be conditionally essential were either present with only one copy (*pgl*, *eda*, *ppc*), or the alternative enzymes could not compensate for their loss (*edd2*, *maeB*, *pdhA2*, alternative FDHs). The degree of essentiality for these genes was correlated to the flux carried by the enzyme (*Figure 6B–D*).

## Discussion

A characteristic feature of all *Burkholderiales* is a fragmented genome organisation (2–4 replicons) (*Fricke et al., 2009*). Comparative genome analysis suggested different evolutionary origins of the *C. necator* chromosomes, with chromosome 1 more conserved among related species than chromosome 2 and pHG1 (*Fricke et al., 2009*). We found that the largest fraction of protein mass (78.7%) can be attributed to chromosome 1, while chromosome 2 and the pHG1 megaplasmid only show strong expression at a few selected loci responsible for alternative lifestyles (lithoautotrophy, denitrification). Chromosome 1 also showed predominantly constitutive expression across different trophic conditions, while the few highly expressed loci on chromosome 2 and pHG1 were transcriptionally regulated. This supports the hypothesis that *C. necator* may have acquired chromosome 2 and pHG1 at a later stage of its evolutionary history and highlights the 'accessory' character of both replicons (*Fricke et al., 2009*).

   Of the 5357 quantified proteins only 1,223 are associated with enzymes and another 93 with central dogma machinery in the *C. necator* RBA model. Yet, utilized enzymes and machinery summed up to 57 % of the protein mass, while 43 % of the proteome was non-utilized, including all proteins not covered by the RBA model. Our estimate for the non-utilized protein mass in *C. necator* is higher than a previously reported estimate for *E. coli* of 26–39 %, particularly regarding the non-modeled protein fraction (39 % in *C. necator* compared to maximally 26 % in *E. coli*) (*O'Brien et al., 2016*). Another estimate for the proportion of non-utilized enzymes for *E. coli* obtained about 30 % of the proteome (*Davidi and Milo, 2017*). We conclude that *C. necator* not only has a larger genome compared to for example *E. coli*, but also expresses many genes without utilizing them in the controlled, homogeneous environments that are typical in biotechnology applications. The large non-utilized protein fraction may be related to environmental readiness and may increase fitness of *C. necator* in the variable and mixed substrate conditions typical of soil (*Hewavitharana et al., 2019*). Further work is necessary to test this hypothesis, for example by subjecting *C. necator* to laboratory evolution experiments in a constant environment with a defined carbon source. Such a selection could lead to inactivation of superfluous substrate assimilation pathways, freeing protein resources and eventually increasing growth rate.

   It is important to note that estimation of protein utilization is not straight-forward and prone to several sources of error. For example, many proteins in *C. necator* are not functionally annotated but

could be catalytically active, eventually leading to underestimation of the utilized protein fraction. On the other hand, enzymes can have 'moonlighting' activities so that the calculated utilization is underestimated for some enzymes and overestimated for others (*Cotton et al., 2020*). Proteins involved in cell motility, cell cycling, sensing of and responding to environmental changes are generally not a part of the metabolic model, yet have vital functions for cellular fitness and are thus utilized in some way. Another challenge is to assign enzyme abundance accurately to reactions that have several annotated proteins, or a protein that is assigned to several enzymatic reactions. In these cases, we divided protein abundance between different enzymes and *vice versa*.

Bearing these limitations in mind, we used the RBA model to investigate the *under*utilization of enzymes. Underutilization as used in this study serves as a proxy for the relation between maximum attainable reaction rate ($V_{max}$) and actual reaction rate, with the latter being shaped by substrate saturation, reverse flux as well as potential allosteric effectors. The estimated enzyme efficiency $k_{app}$ is influenced by these factors and can deviate from in vitro measured maximum turnover $k_{cat}$ (*Davidi et al., 2016*). A general observation regarding utilization is the dependency on growth rate. Flux of metabolic enzymes is directly proportional to growth rate, given that all other cultivation parameters are kept constant. At low growth and low flux through metabolism, bacteria optimize fitness by reallocating protein resources from growth functions (ribosomes) to substrate assimilation (transporters) (*Scott et al., 2014*; *Hui et al., 2015*; *Jahn et al., 2018*). However, this reallocation is only a gradual response and neither results in full reduction of superfluous proteome sectors, nor the shrinking of the protein pool (g protein/gDCW). The consequence is that enzyme utilization becomes low at low growth rates (*O'Brien et al., 2016*). *C. necator* also shows this pattern: ribosomal proteins are incompletely reduced at low growth rates, and enzymes of central metabolism generally remain highly abundant (*Figure 3—figure supplement 1*, *Figure 4*), effectively creating an underutilized enzyme reserve.

Underutilization of enzymes represents an 'efficiency sacrifice' for host fitness. Expression of excess non-metabolic proteins such as LacZ or YFP reduces bacterial growth rate (*Hui et al., 2015*; *Jahn et al., 2018*). However, several recent experimental studies have shown that enzyme underutilization in *E. coli* central metabolism, such as in the OPP pathway and amino acid biosynthesis, provides a buffer against perturbations in environmental conditions or gene expression (*Davidi and Milo, 2017*; *Christodoulou et al., 2018*; *Sander et al., 2019*). The importance of underutilized enzymes for metabolic stability has also been shown for metabolic networks such as the CBB cycle (*Barenholz et al., 2017*; *Janasch et al., 2018*). We observed that highly abundant enzymes are better utilized and less variable across conditions. This is most likely a result of the evolutionary pressure on enzyme reserve costs, which increase proportionally with the abundance of enzymes.

It is of interest to compare enzyme utilization in *C. necator* to *E. coli,* a model bacterium with a different environmental niche. The central carbon metabolism pathways of *C. necator* showed differences in enzyme abundance, variability, and utilization. Abundance of enzymes for the upper EMP pathway, PPP, and CBB cycle was on average higher than for the enzymes of the ED pathway, pyruvate metabolism or TCA. This is similar to *E. coli*, where higher abundance of glycolysis enzymes was explained by high flux demand and low thermodynamic driving force (*Noor et al., 2016*). But enzymes of the upper EMP pathway and PPP also showed strong transcriptional regulation (variability in gene expression, *Figure 3D*), which is a marked difference to *E. coli*, where enzyme levels show low variation across multiple growth conditions (*Schmidt et al., 2016*), and flux is mainly regulated through allosteric interactions (*Reznik et al., 2017*). Of all central carbon metabolism, the TCA cycle enzymes showed on average lowest abundance, variability and -for most enzymes- utilization. This is similar to *E. coli*, where a simple enzyme cost minimization model suggested lower enzyme abundance than what was measured experimentally (*Noor et al., 2016*). Only when reverse fluxes (for reactions with low thermodynamic driving force) and low enzyme saturation ([S]< $K_M$, estimated from metabolite levels), were taken into account, was the calculated enzyme demand similar to the measured levels (*Noor et al., 2016*). The RBA framework does not take thermodynamic driving forces into account and may therefore underestimate enzyme demand for such reactions.

How was the regulatory network in *C. necator*'s central carbon metabolism shaped by its native environment? *E. coli* is adapted to regular nutrient upshifts every 2–3 hr (*Mori et al., 2017*). It therefore evolved allosteric regulation to deal with quickly changing fluxes through the EMP pathway, its prime catabolic route (*Reznik et al., 2017*). For *C. necator*, sugars are likely not the preferred

substrate as the only sugars it utilizes are fructose and N-acetylglucosamine (*Cramm, 2009*). Flux through the upper EMP pathway is low as it uses the low-yield ED pathway to catabolize sugars. A slow but more resource efficient transcriptional regulation of glycolysis could therefore provide a fitness benefit for an environment with limited and irregular substrate supply. Interestingly, only the glycolysis/PPP enzymes located on the phylogenetically young *cbb* operons are transcriptionally regulated, while the ancestral enzymes on chromosome 1 are constitutively expressed (*Figure 4— figure supplement 1*). These enzymes are also scattered over the chromosome and therefore not collectively regulated. The diverging regulation for glycolysis-related genes could mark a branching point in the evolutionary history of *C. necator*. The pHG1 plasmid was likely acquired recently, based on its transmissibility and proven ability to confer hydrogenotrophic metabolism (*Friedrich et al., 1981*). *Cbb* genes could either get lost or take over the function as main glycolysis enzymes from their chromosome 1 orthologs.

The two copies of the *cbb* operon in *C. necator* are of hybrid nature as CBB cycle enzymes functionally overlap with EMP glycolysis and PPP. Expression of the *cbb* operon depended on the supplied substrate and was highest for growth on formate, where CBB cycle genes are essential. However, a more complex picture emerged for *cbb* expression during other substrate limitations (increasing with μ on fructose, decreasing with μ on succinate). The *cbb* operon is transcriptionally regulated by two systems, CbbR (*Bowien and Kusian, 2002*) and RegA/B (*Gruber et al., 2017*). RegA/B guarantees a basic level of constitutive expression, while CbbR senses the intracellular PEP concentration (*Gruber et al., 2017*). PEP is an important allosteric regulator responsible for the switch between glycolytic and gluconeogenic flux in *E. coli* (*Reznik et al., 2017*). In *C. necator*, growth on fructose leads to low PEP concentration, triggering *cbb* expression, while it is the other way around for succinate. This prompts the question which evolutionary benefit cells gain from *cbb* expression during heterotrophic growth? On substrates with a higher degree of reduction than biomass, such as glycerol, there will be sufficient redox power to fix emitted $CO_2$ (*Alagesan et al., 2017*; *Guadalupe-Medina et al., 2013*). On substrates with a lower degree of reduction, such an excess is not expected. It has also been shown that reassimilation of emitted $CO_2$ by Rubisco improves PHB yield (*Shimizu et al., 2015*). We generalized this hypothesis and tested if CBB activity could also provide a biomass yield or growth benefit. Our model simulations suggested that $CO_2$-reassimilation is unlikely to provide such a benefit as long as there is no additional energy source (Rubisco activity even causes a higher net $CO_2$ emission). Down-regulation of the *cbb* operon (*cbbR* mutant) caused no significant fitness change on fructose or succinate, suggesting that $CO_2$ fixation in these conditions provides no benefit. The resolution of the transposon library experiments was however too low to exclude that CBB activity does not confer a small growth advantage. We propose that the conserved PEP-dependent transcriptional regulation of *cbb* leads to a collateral expression of Rubisco in conditions where it is not required, such as fructose. This is a remarkable example of suboptimality, where one benefit could be readiness for lithoautotrophic growth when hydrogen or formate become available. Probing the effect of *cbb* gene knockouts with the transposon library also revealed that *C. necator* can compensate the loss of any *cbb* gene by expressing the respective second copy. This finding applies to central carbon metabolism in general. Almost all enzyme functions are covered by several gene loci, so that knockout did not result in fitness loss. Notable exceptions are the reactions of the ED pathway, PEP carboxylase (*ppc*), and malic enzyme (*maeA*), that showed significantly reduced fitness in conditions where these reactions carry high flux.

Our results highlight the metabolic flexibility of *C. necator* and its robustness to changing environmental conditions. Its high degree of genomic redundancy makes it tolerant to gene loss, but may also lead to regulatory conflicts exemplified by *cbb* expression. A comparison of microbial genomes showed that the CBB cycle is accompanied by a metabolism-wide range of adaptations (*Asplund-Samuelsson and Hudson, 2021*). Considering a possibly recent acquisition of the CBB cycle *via* pHG1, it is likely that *C. necator* is currently evolving to make best use of the *cbb* genes. Our results also imply that *C. neactor* is in its current state far from being an ideal host for biotech applications. This is because (1) gene duplications and iso-enzymes complicate genetic engineering, (2) expression of unutilized pathways is protein-inefficient, (3) a large pool of uncharacterized enzymes makes it difficult to control metabolic flux (*Figure 2*). Strategies to tackle these problems could include both targeted and untargeted approaches. The systematic deletion of undesired functions could result in higher enzyme efficiency and therefore higher product yield. One example is the removal of costly

hydrogenase expression for growth on formate. Alternatively, laboratory evolution could be employed to select mutants with beneficial traits such as tolerance to formic acid.

# Materials and methods

## Key resources table

| Reagent type (species) or resource | Designation | Source or reference | Identifiers | Additional information |
|---|---|---|---|---|
| Strain, strain background (*Cupriavidus necator*) | H16 (wild type) | German Collection of Microorganisms and Cell Cultures, DSM-428 | NCBI:txid381666 | https://www.dsmz.de/collection/catalogue/details/culture/DSM-428 |
| Strain, strain background (*Cupriavidus necator*) | H16 PHB⁻4(mutant deficient in PHB synthesis) | German Collection of Microorganisms and Cell Cultures, DSM-541 | H16 PHB⁻4 | https://www.dsmz.de/collection/catalogue/details/culture/DSM-541 |
| Strain, strain background (*Cupriavidus necator*) | H16, transposon mutant library (60,000 individual mutants) | This study | NCBI:txid381666 | Obtained by conjugation with *E. coli* APA766 |
| Strain, strain background (*Escherichia coli*) | APA766, transposon donor strain (pKMW7 Tn5) | *Wetmore et al., 2015* | WM3064 | Obtained from the original authors (Adam Deutsch- bauer lab) |

## Strains and cultivation

*Cupriavidus necator* H16 was obtained from the German Collection of Microorganisms and Cell Cultures, strain number DSM-428. Cells were cultivated on complete (LB) medium, or minimal medium depending on experimental setup. Minimal medium was composed of 0.78 g/L $NaH_2PO_4$, 4.18 g/L $Na_2HPO_4 \times 2H_2O$, 1 g/L $NH_4Cl$, 0.1 g/L $K_2SO_4$, 0.1 g/L $MgCl_2 \times 6H_2O$, 1.6 mg/L $FeCl_3 \times 6H_2O$, 0.4 mg/L $CaCl_2$, 0.05 mg/L $CoCl_2 \times 6H_2O$, 1.8 mg/L $Na_2MoO_4 \times 2H_2O$, 0.13 g/L $Ni_2SO_4 \times 6H_2O$, 0.07 mg/L $CuCl_2 \times 2H_2O$. Depending on the experiment, 0.5 g/L D-fructose, 0.5 g/L succinate, or 1.5 g/L pH-neutralized formic acid was added as carbon source. For nitrogen limitation, the concentration of D-fructose was increased to 2 g/L and concentration of $NH_4Cl$ was reduced to 0.025 g/L. All components were added to autoclaved sodium phosphate buffer from filter-sterilized stock solutions. Batch cultures were grown in 100 mL shake flasks at 30 °C and 180 RPM. Precultures of the barcoded *C. necator* transposon library were supplemented with 200 µg/mL kanamycin and 50 µg/mL gentamicin to suppress growth of untransformed *C. necator* recipient or *E. coli* donor cells.

## Chemostat bioreactors

*C. necator* H16 (wild type) or the *C. necator* H16 transposon mutant library was cultivated in an 8-tube MC-1000-OD bioreactor (Photon System Instruments, Drasov, CZ). The system was customized to perform chemostat cultivation as described previously (*Jahn et al., 2018*; *Yao et al., 2020*). Bioreactors (65 mL) were filled with minimal medium supplemented with the respective carbon and nitrogen source, and inoculated with an overnight preculture to a target $OD_{720nm}$ of 0.05. Bioreactors were bubbled with air at a rate of 12.5 mL/min and a temperature of 30 °C. The $OD_{720nm}$ and $OD_{680nm}$ were measured every 15 min. Fresh medium was continuously added using Reglo ICC precision peristaltic pumps (Ismatec, GER). For pulsed chemostat experiments, a volume corresponding to the continuous addition of medium over a given time period was added in a single pulse every 2 hr. For proteomics, 40 mL samples were taken after five retention times of continuous growth at a fixed dilution rate ($t_R$ = 1/D; for example $t_R$(D = 0.1 h⁻¹) = 1 / 0.1 = 10 hr). For transposon library competition experiments, 15 mL samples were taken after 0, 8, and 16 generations of growth (population average). Cells were harvested by centrifugation for 10 min at 5000 xg, 4 °C, washed with 1 mL ice-cold PBS, transferred to a 1.5 mL tube, and centrifuged again for 2 min at 8000 xg, 4 °C. The supernatant was discarded and the pellet frozen at –20 °C.

## Determination of biomass yield

Substrate uptake rate $q_S$ was determined using the dilution rate D, the culture volume V, the biomass concentration $c_{bm}$ in gDCW L⁻¹, and the initial and residual substrate concentrations $S_i$ and $S_r$,

respectively, in the following equation: $q_S = \frac{V \cdot D \cdot (S_i - S_r)}{c_{bm}}$ . The biomass yield $Y_{X/S}$ for all substrates was determined by fitting a linear model to the growth rate-substrate uptake rate relationship.

## Dry cell weight determination

Dry cell weight (DCW) measurements for carbon limitation were carried out in shake flasks. Fifty mL of minimal medium were supplemented with 0.5 g/L fructose, 0.5 g/L succinate, or 2 g/L formate (pH neutralized). Flasks were inoculated with *C. necator* to an $OD_{600}$ of 0.01 and cultivated for 48 hr at 30 °C before harvesting. DCW measurement for nitrogen limitation was carried out using an ammonium limited chemostat as described above (2 g/L fructose, 0.05 g/L $NH_4Cl$). A total of 50 mL cell suspension was harvested by centrifugation for 10 min, 5000 xg, 4 °C. The pellet was washed twice with 1 mL mqH2O, transferred to preweighed 1.5 mL tubes and dried for 4 hr at 70 °C. Dried cell mass was measured on a precision scale. Biomass yield for formate batch cultures was corrected using the linear relationship of yield reduction and residual formate concentration from *Grunwald et al., 2015*.

## Determination of PHB content

Pellets from DCW determination were dissolved in 1 ml of sodium hypochlorite solution (10–15 % chlorine) and incubated at 37 °C for 1 hr for cell lysis. The lysate was harvested by centrifugation at 16,000 xg for 2 min, RT. The pellet was sequentially washed with 1 mL mqH2O, 1 ml acetone, and 1 ml of 96 % ethanol. The lysate was harvested by centrifugation at 16,000 xg for 2 min, ethanol was completely removed, and the pellet resuspended in 1 mL chloroform. The solution was transferred to a 5 mL glass tube and heated for 2 min at 70 °C to extract PHB. The solution was then cooled to RT and centrifuged for 2 min at 4000 xg. The supernatant was transferred to a fresh glass tube. The PHB extraction of the pellet was repeated with one additional mL chloroform and the samples were pooled. The chloroform was evaporated completely at 40°C to 50°C overnight in a vented hood. For hydrolysis of PHB into crotonic acid, 1 ml of concentrated sulphuric acid was added to the precipitate and samples were incubated at 100 °C for 10–20 min. The hydrolysate was diluted 1:100 by mixing 10 μL sample with 990 μL 14 mM $H_2SO_4$. For each sample, 3 × 100 μL were transferred to a low-UV-absorption 96-well plate and UV absorbance of crotonic acid was measured at 235 nm in a spectrophotometer. For PHB quantification, absorption was compared to a standard curve of PHB hydrolysate with known concentration. For the standard, 10 mg of pure PHB were hydrolysed in concentrated H2SO4 as described above. The standard was diluted 1:10 by mixing 500 μL with 4.5 mL 14 mM $H_2SO_4$ resulting in a 1 mg/mL stock solution. Dilutions ranging from 0.0 to 1.0 mg/mL were measured in a 96-well plate as described above.

## Residual substrate measurement with HPLC

Culture supernatant was obtained after centrifugation of cell samples. A volume of 1 mL supernatant was transferred to an LC glass vial using Millex-HV PVDF syringe filter tips (Merck Millipore). The HPLC column (Aminex 300 mm HPX-87H) was equilibrated with 5 mM $H_2SO_4$ as mobile phase for 1 hr, at a flow rate of 0.5 mL/min. The column was heated to 60 °C. A volume of 20 μL per sample was injected to the HPLC followed by a run time of 30 min. UV-absorption was constantly detected at 210 nm wavelength. Standards with four different concentrations, 10, 50, 100, and 200 mg/L, were used for quantification of each residual substrate (succinate, formate, fructose, ammonium chloride). Calibration curves were obtained by fitting a linear equation to the concentration-absorbance relationship. Residual substrate concentration was then determined by solving the equation with the obtained absorbance measurements.

## Statistical analysis

Bioreactor cultivations, LC-MS/MS measurement for proteomics, and library competition experiments ('BarSeq') were performed with four independent biological replicates. HPLC measurement of supernatants was performed with three biological replicates. Here, biological replicate means that samples were obtained from independently replicated bioreactor cultivations inoculated with the same preculture. The sample size of four was chosen based on the known variance from previous proteomics experiments. If not otherwise indicated in figure legends, points and error bars represent the mean and standard deviation. No removal of outliers was performed. All analyses of proteomics,

modeling, and fitness data are documented in R notebooks available at https://github.com/m-jahn/R-notebook-ralstonia-proteome.

## Sample preparation for LC-MS/MS

Frozen cell pellets were resuspended in 125 µL solubilization buffer (200 mM TEAB, 8 M Urea, protease inhibitor). A total of 100 µL glass beads (100 µm diameter) were added to the cell suspension and cells were lysed by bead beating in a Qiagen TissueLyzer II (5 min, f = 30 /s, precooled cassettes). Cell debris was removed by centrifugation at 14,000 xg, 30 min, 4 °C, and supernatant was transferred to a new tube. Protein concentration was determined using the Bradford assay (Bio-Rad). For reduction and alkylation of proteins, 2.5 µL 200 mM DTT (5 mM final) and 5 µL 200 mM CAA (10 mM final) were added, respectively, and samples incubated for 60 min at RT in the dark. Samples were diluted 8 x with 700 µL 200 µM TEAB. For digestion, Lys-C was added in a ratio of 1:75 w/w to protein concentration, and samples were incubated at 37 °C and 600 RPM for 12 hr. Trypsin was added (1:75 w/w) and samples incubated for 24 hr at the same conditions. Samples were acidified with 100 µL 10 % formic acid (FA) and insoluble compounds were removed by centrifugation (14,000 xg, 15 min, RT). Peptide samples were then cleaned up using a solid phase extraction (SPE) protocol in 96-well plate format (Tecan Resolvex A200) according to the manufacturer's recommendations. Briefly, the 96-well plate with SPE inserts was equilibrated with 200 µL acetonitrile (ACN) and 2 × 200 µL 0.6 % acetic acid. A lysate volume corresponding to 40 µg protein was loaded on the plate and washed twice with 200 µL 0.6 % acetic acid. Peptides were eluted from the column in 100 µL elution buffer (0.6 % acetic acid, 80 % ACN) and dried in a speedvac for 2 hr, 37 °C. Dried peptides were frozen at –80 °C and dissolved in 10 % FA to a final concentration of 1 µg/µL before MS measurement.

## LC-MS/MS analysis of lysates

Lysates were analyzed using a Thermo Fisher Q Exactive HF mass spectrometer (MS) coupled to a Dionex UltiMate 3,000 UHPLC system (Thermo Fisher). The UHPLC was equipped with a trap column (Acclaim PepMap 100, 75 µm x 2 cm, C18, P/N 164535, Thermo Fisher Scientific) and a 50 cm analytical column (Acclaim PepMap 100, 75 µm x 50 cm, C18, P/N ES803, Thermo Fisher Scientific). The injection volume was 2 µL out of 18 µL in which the samples were dissolved in the autosampler. Chromatography was performed using solvent A (3 % ACN, 0.1 % FA) and solvent B (95 % ACN, 0.1 % FA) as the mobile phases. The peptides were eluted from the UHPLC system over 90 min at a flow rate of 250 nL/min with the following mobile phase gradient: 2 % solvent B for 4 min, 2–4 % solvent B for 1 min, 4–45 % solvent B for 90 min, 45–99 % solvent B for 3 min, 99 % solvent B for 10 min and 99–2 % solvent B for 1 min following re-equilibration of the column at 2 % solvent B for 6 min. The MS was operated in a data-dependent acquisition mode with a Top eight method. The MS was configured to perform a survey scan from 300 to 2000 m/z with resolution of 120,000, AGC target of $1 \times 10^6$, maximum IT of 250 ms and eight subsequent MS/MS scans at 30,000 resolution with an isolation window of 2.0 m/z, AGC target of $2 \times 10^5$, maximum IT of 150 ms and dynamic exclusion set to 20 s.

## Protein identification and quantification

Thermo raw spectra files were converted to the mzML standard using Proteowizard's MSConvert tool. Peptide identification and label-free quantification were performed using OpenMS 2.4.0 in KNIME (*Röst et al., 2016*; RRID:SCR_006164). The KNIME pipeline for MS data processing was deposited on https://github.com/m-jahn/openMS-workflows (labelfree_MSGFplus_Percolator_FFI.knwf). MS/MS spectra were subjected to sequence database searching using the OpenMS implementation of MS-GF+ and Percolator (*Granholm et al., 2014*) with the *Cupriavidus necator* H16 reference proteome as database (NCBI assembly GCA_000009285.2, downloaded 07 January 2019). Carbamidomethylation was considered as a fixed modification on cysteine and oxidation as a variable modification on methionine. The precursor ion mass window tolerance was set to 10 ppm. The PeptideIndexer module was used to annotate peptide hits with their corresponding target or decoy status, PSMFeatureExtractor was used to annotate additional characteristics to features, PercolatorAdapter was used to estimate the false discovery rate (FDR), and IDFilter was used to keep only peptides with q-values lower than 0.01 (1 % FDR). The quantification pipeline is based on the FeatureFinderIdentification workflow allowing feature propagation between different runs (*Weisser and Choudhary, 2017*). MzML files were retention time corrected using MapRTTransformer, and identifications (idXML files)

were combined using the IDMerger module. FeatureFinderIdentification was then used to generate featureXML files based on all identifications combined from different runs. Individual feature maps were combined to a consensus feature map using FeatureLinkerUnlabelledKD, and global intensity was normalized using ConsensusMapNormalizer (by median). Protein quantity was determined by summing up the intensities of all unique peptides per protein. Abundance of ambiguous peptides (peptides mapping to two different proteins) were shared between proteins.

## Creation of barcoded *C. necator* transposon library

The transposon library was prepared according to the RB-TnSeq workflow described in *Wetmore et al., 2015*. Briefly, *C. necator* H16 wild type was conjugated with an *E. coli* APA766 donor strain containing a barcoded transposon library. The strain is auxotrophic for DAP, the L-Lysin precursor 2,6-diamino-pimelate, to allow for counter selection. Overnight cultures of *E. coli* APA766 and *C. necator* H16 in 10 mL LB medium in shake flasks were prepared. The APA766 culture was supplemented with 0.4 mM DAP and 50 µg/mL kanamycin. 2 L of LB medium (APA766 with 0.4 mM DAP and 50 µg/mL kanamycin) in shake flasks was each inoculated with the respective pre-cultures and incubated overnight at 30 °C and 180 RPM. Cells were harvested during exponential growth phase by centrifugation for 10 min, 5000 xg, RT. Supernatant was discarded, cell pellets were resuspended in residual liquid, transferred to 2 mL tubes, washed twice with 2 mL PBS, and finally resuspended in a total amount of 500 µL PBS. Cell suspensions from both strains were combined and plated on 25 cm x 25 cm large trays (Q-tray, Molecular Devices) with LB agar supplemented with 0.4 mM DAP. For conjugation, plates were incubated overnight at 30 °C. Cells were then harvested from mating plates by rinsing with 200 µL PBS. The cell suspension was plated on selection plates with LB agar supplemented with 100 µg/mL kanamycin, without DAP. After colonies of sufficient size appeared, transformants were harvested by scraping all cell mass from the plate and collecting the pooled scrapings in 1.5 mL tubes. The mutant library diluted tenfold and was then immediately frozen at –80 °C. For competition experiments, a 1 mL 10-fold diluted aliquot (pool of all conjugations, ~ 1 M CFU) was used to inoculate pre-cultures.

## Mapping of transposon mutants (TnSeq)

A 1 mL aliquot of the diluted pooled library scrapings was used to inoculate a 50 mL LB culture (with 200 µg/mL kanamycin) and grown overnight at 30 °C, 200 RPM. DNA was extracted from 1 mL of this outgrown culture using a GeneJet Genomic DNA Purification Kit (ThermoScientific) and the concentration of genomic DNA was quantified using a Qubit dsDNA HS Assay Kit (Invitrogen). A 1 µg aliquot of genomic DNA was suspended in 15 µL of 10 mM Tris buffer, placed in a microTUBE-15 AFA Beads tube (Covaris) and fragmented into 300 bp fragments using an ME220 focused ultrasonicator (Covaris) with waveguide 500,526 installed. Cycle time was increased to 60 s, all other settings were taken from manufacturer's recommendation for generating 350 bp fragments. Fragment end repair and adaptor ligation was performed using an NEBNext Ultra II DNA Library Prep Kit (New England Biolabs) following the manufacturer's protocol. Size selection of NEB adaptor ligated fragments was carried out using SPRISelect magnetic beads (Beckman Coulter) following the method in the NEBNext Ultra II DNA Library Prep Kit User manual. To enrich transposon-containing sequences, a 30 cycle PCR amplification was performed using the Biotin_Short_pHIMAR and NC102 primers (*Supplementary file 3*) using Q5 mastermix (New England Biolabs). Cycle conditions were 30 s 98 °C followed by 30 cycles (15 s 98 °C, 75 s 72 °C) and a 5 min 72 °C final extension. The biotinylated transposon-containing sequences were purified using MyOne Streptavidin T1 Dynabeads (Invitrogen) according to the manufacturer's instructions. The transposon containing DNA was then stripped from the beads by resuspending the beads in 25 µL of MilliQ water followed by incubation at 70 °C for 10 min. The beads were separated by incubation on a magnetic stand at room temperature for 1 min and the supernatant was recovered. Adaptors for Illumina sequencing were added via PCR amplification using Nspacer_barseq_pHIMAR (*Wetmore et al., 2015*) and NEBNext Index 3 Primer for Illumina (New England Biolabs). Cycle conditions were 30 s 98 °C followed by four cycles (15 s 98 °C, 75 s 72 °C) and a 5 min 72 °C final extension. PCR products were separated on a 1 % agarose gel and gel extraction was performed on the band between 300–600 bp using a Gel Extraction Kit (ThermoScientific). The DNA concentration of the samples were quantified using a Qubit dsDNA HS Assay Kit (Invitrogen) and diluted to 2 nM. The 2 nM library was diluted, denatured and sequenced using a NextSeq 500/550

Mid Output Kit v2.5 150 Cycles, (Illumina) run on a NextSeq 550 instrument (Illumina) according to the manufacturer's instructions. Library loading concentration was 1.8 pM with a 10 % phiX spike. Reads containing barcodes and genomic DNA fragments were mapped to the *C. necator* genome following the protocol from *Wetmore et al., 2015*. Briefly, the scripts *MapTnSeq.pl* and *DesignRandomPool. pl* from https://bitbucket.org/berkeleylab/feba/src/master/ (*Price, 2021*) were adapted to map reads to the reference genome, and to summarize read counts per barcode, respectively. Only barcodes mapping to the same region with at least two reads were included. The automatic pipeline for TnSeq data analysis is available at https://github.com/m-jahn/TnSeq-pipe.

## Gene essentiality analysis

TnSeq data from two different iterations of the transposon library were combined to obtain high insertion frequency per gene (72,443 and 57,040 mutants, respectively). Of the 129,483 transposon insertions, 23,339 mapped to intergenic regions and were excluded from essentiality analysis. Of all insertions mapping to a gene, 78.7 % were localized within the central 80 % of the ORF and were considered as true knockouts. Following the method from *Rubin et al., 2015*, a metric for essentiality was calculated, the insertion index *II*; *II* is the number of transposon insertions *n* of a gene *i* with length *k* divided by insertions per region *r* (average of 100 genes around the target position):

$$II_i \; = \; \left( n_i \, / \, k_i \right) \, / \, \left( n_r \, / \, k_r \right)$$

The *II* is bimodally distributed, one set of genes is hit by transposons at an average rate while other genes are hit with lower frequency. To determine an *II* threshold for essentiality, two gamma distributions were fitted to the assumed populations of (1) essential and (2) non-essential genes. For all possible *II*, the probability of falling into the essential and non-essential distribution was determined and a fivefold difference defined as lower and upper thresholds to count a gene as essential or non-essential, respectively. Genes with *II* between the two thresholds were flagged as ambiguous (*p* denotes the probability density function of *II* for essential and non-essential genes):

$$II_{ambiguous} \; = \; II \, \left[ \left( p_{ess} \; < \; 5 \cdot p_{non-ess} \right) \; \cup \; \left( p_{non-ess} \; < \; 5 \cdot p_{ess} \right) \right]$$

To estimate essentiality of enzymes/reactions instead of genes, each enzyme with at least one associated gene being essential was counted as essential, and each enzyme associated with at least one probably essential gene was counted as probably essential; all other enzymes were marked as non-essential.

## Gene fitness analysis (BarSeq)

Frozen cell pellets from the pulsed and continuous competition experiments were resuspended in 100 µL of 10 mM Tris and genomic DNA was extracted from 10 µL of the resuspension using a GeneJet Genomic DNA Purification Kit (ThermoScientific). Amplification of the barcodes from genomic DNA was conducted using one of the custom forward indexing primers (BarSeq_F_i7_001 - BarSeq_F_i7_036, *Supplementary file 3*) and the reverse phasing primer pool (BarSeq_R_P2_UMI_ Univ - BarSeq_R_P2_UMI_Univ_N5). For each sample, 9 µL of genomic DNA extract ( ≥ 10 ng/µL) was combined with 3 µL of a forward indexing primer (100 nM), 3 µL of the reverse phasing primer pool (100 nM) and 15 µL of Q5 Mastermix (New England Biolabs). Cycle conditions were 4 min at 98 °C followed by 20 x (30 s at 98 °C, 30 s at 68 °C and 30 s at 72 °C) with a final extension of 5 min at 72 °C. Concentrations of each sample was quantified using a Qubit dsDNA HS Assay Kit (Invitrogen). Samples were then pooled with 40 ng from up to 36 different samples being combined and run on a 1 % agarose gel. Gel extraction was performed on the thick band centered around 200 bp and the concentration of the purified pooled library was quantified again via Qubit assay and diluted down to 2 nM. The 2 nM library was then diluted, denatured and sequenced using a NextSeq 500/550 High Output Kit v2.5 (75 Cycles) (Illumina) run on a NextSeq 550 instrument (Illumina) according to the manufacturer's instructions. Library loading concentration was 1.8 pM with a 1 % phiX spike. Gene fitness was calculated from read counts per barcoded mutant based on the method from *Wetmore et al., 2015*. Briefly, scripts from https://bitbucket.org/berkeleylab/feba/src/master/ were adapted to trim and filter reads, extract barcodes, and summarize read counts per barcode. Fitness score

calculation based on the $log_2$ fold change of read count per barcode over time was implemented as an R script. The automatic pipeline for BarSeq analysis is available at https://github.com/Asplund-Samuelsson/rebar. Altogether, fitness for 5441 genes was quantified with an average of 6.4 insertion mutants per gene. The remaining 1173 genes were either essential (no viable insertion mutant), probably essential (number of transposon mutants in the surrounding region too low to determine essentiality), or fitness could not be quantified with sufficient confidence (low read count). A significance threshold of $|F| \geq 3$ after at least eight generations was chosen based on the bulk fitness distribution of mutants ($-2 \leq F \leq 2$).

## Resource balance analysis model

The resource balance analysis (RBA) model for *C. necator* H16 was generated using the RBApy package (*Bulović et al., 2019*). The model and a detailed description of its generation is available at https://github.com/m-jahn/Bacterial-RBA-models/. The main input was the curated genome scale model for *C. necator* in SBML format (1360 reactions, excluding exchange reactions), available at https://github.com/m-jahn/genome-scale-models. Amino acid sequence, subunit stoichiometry and cofactor requirements for all proteins associated with model reactions were automatically retrieved from uniprot (organism ID: 381666). Fasta files detailing the composition of the ribosome (3 rRNA and 68 proteins), chaperones (eight proteins), DNA polymerase III (eight proteins), and RNA polymerase II (nine proteins) were added manually. Rates for these macromolecular 'machines' were adopted from published values for *E. coli* (*Supplementary file 1*). Rates for ribosome and chaperone were taken from *Bulović et al., 2019*, rate of RNA polymerase was taken from *Epshtein and Nudler, 2003*, and rate of DNA polymerase was the average of several published values obtained from https://bionumbers.hms.harvard.edu (IDs 102052, 104938, 109251, 111770). Biomass composition of *C. necator* H16, growth- and non-growth associated maintenance were all taken from *Park et al., 2011*. A growth rate dependent flux towards PHB was added (3 mmol gDCW$^{-1}$) to obtain biomass yields corresponding to experimentally determined values. The model was calibrated by adding estimates for $k_{app}$, the apparent catalytic rate for each metabolic enzyme, following the procedure in *Bulović et al., 2019*. For each model reaction and substrate limitation, flux boundaries were obtained from flux sampling analysis (FSA) using COBRApy (*Ebrahim et al., 2013*), and enzyme abundance in mmol gDCW$^{-1}$ was obtained from proteomics measurements. $k_{app}$ was determined by calculating the maximum flux per unit enzyme over all conditions. For enzymes without estimated $k_{app}$ (no flux, or no protein abundance available), the median of the $k_{app}$ distribution was used (5770 hr$^{-1}$, *Figure 2—figure supplement 1*). The average protein fraction of cell dry weight was taken from *Park et al., 2011*. The reported concentration of 0.68 g protein gDCW$^{-1}$ was converted to mmol amino acids gDCW$^{-1}$ by assuming an average molecular weight per amino acid of 110 g mol$^{-1}$:

$$c = \frac{0.68 \ g \cdot mol \cdot 1000}{gDCW \cdot 110 \ g} = 6.18 \ mmol \ gDCW^{-1}$$

Proteome fraction per cellular compartment (cytoplasm, cytoplasmic membrane) was estimated based on proteomics measurements and predicted protein localization (psortb algorithm) as input. Growth-rate-dependent fractions for cytoplasmic and membrane proteins were obtained by correlating growth rate and the respective mass fractions and fitting a linear model (*Figure 2—figure supplement 1*). The same procedure was applied to estimate the non-enzymatic protein fraction per compartment. Proteins not contained in the model were categorized as non-enzymatic as they have no catalytic function in the model (*Supplementary file 1*, *Figure 2—figure supplement 1*).

## Data and software availability

The mass spectrometry proteomics data have been deposited to the ProteomeXchange Consortium via the PRIDE partner repository with the dataset identifier PXD024819. Protein quantification results can be browsed and interactively analyzed using the web application available at https://m-jahn.shinyapps.io/ShinyProt. All sequencing data for TnSeq and BarSeq experiments are available at the European Nucleotide Archive with accession number PRJEB43757. The data for competition experiments performed with the transposon mutant library can be browsed and interactively analyzed using the web application available at https://m-jahn.shinyapps.io/ShinyLib/.

The openMS/KNIME workflow for MS data processing is available at https://github.com/m-jahn/openMS-workflows, (copy archived at swh:1:rev:dd4e0a20a39300cac9ad89840862348895e9f907, *Jahn, 2021c*). The revised genome scale model of *C. necator* H16 is available at https://github.com/m-jahn/genome-scale-models, (copy archived at swh:1:rev:d2cdcdfbdf140694993a3108b1a10715566f09aa, *Jahn, 2021b*). The resource balance analysis (RBA) model of *C. necator* H16 is available at https://github.com/m-jahn/Bacterial-RBA-models, (copy archived at swh:1:rev:efe12f7d53810e1ae-c618b2b2da0fa8a49aec1c5, *Jahn, 2021a*). The code used to process TnSeq data from raw fastq files (read trimming, filtering, mapping to genome) is available at https://github.com/m-jahn/TnSeq-pipe, (copy archived at swh:1:rev:1f256a366f772a2450c2bcfecc43bb2181efc989, *Jahn, 2021e*). The code used to process BarSeq data from raw fastq files is available at https://github.com/Asplund-Samuelsson/rebar, (copy archived at swh:1:rev:5a4dbad30041bc0510a9a2eed55b1a47f705ff51, *Asplund-Samuelsson, 2021*). All analyses of proteomics, modeling, and fitness data were performed using the R programming language and are documented in R notebooks available at https://github.com/m-jahn/R-notebook-ralstonia-proteome, (copy archived at swh:1:rev:fde3cf9f8a6ea05d2d-ba30606d64c13867e0557a, *Jahn, 2021d*).

## Acknowledgements

We acknowledge Julia Foyer and Arvid Gynnå for assistance in bioreactor cultivations. We like to thank Anne Goelzer for guidance with resource balance analysis modeling. This study was financially supported by the Swedish Research Council Vetenskapsrådet (Grant number 2016–06160), the Swedish Research Council Formas (Grant number 2015–939 and 2019–01491), and Novo Nordisk Fonden (Grant number NNF20OC0061469).

## Additional information

### Funding

| Funder | Grant reference number | Author |
| --- | --- | --- |
| Vetenskapsrådet | 2016-06160 | Nick Crang Kyle Kimler Johannes Asplund-Samuelsson |
| Swedish Research Council Formas | 2019-01491 | Michael Jahn |
| Swedish Research Council Formas | 2015-939 | Michael Jahn |
| Novo Nordisk Fonden | NNF20OC0061469 | Markus Janasch Elton Paul Hudson |

The funders had no role in study design, data collection and interpretation, or the decision to submit the work for publication.

### Author contributions

Michael Jahn, Conceptualization, Data curation, Formal analysis, Funding acquisition, Investigation, Methodology, Project administration, Resources, Software, Visualization, Writing - original draft, Writing - review and editing; Nick Crang, Investigation, Methodology, Resources, Writing - review and editing; Markus Janasch, Conceptualization, Methodology, Writing - review and editing; Andreas Hober, Björn Forsström, Investigation, Methodology, Resources; Kyle Kimler, Data curation, Investigation, Methodology, Resources; Alexander Mattausch, Investigation, Resources, Visualization; Qi Chen, Formal analysis, Methodology, Software; Johannes Asplund-Samuelsson, Data curation, Formal analysis, Methodology, Software, Writing - review and editing; Elton Paul Hudson, Conceptualization, Funding acquisition, Project administration, Supervision, Writing - original draft, Writing - review and editing

Author ORCIDs

Michael Jahn http://orcid.org/0000-0002-3913-153X
Nick Crang http://orcid.org/0000-0002-7569-6597
Markus Janasch http://orcid.org/0000-0001-7745-720X
Andreas Hober http://orcid.org/0000-0001-8947-2562
Björn Forsström http://orcid.org/0000-0002-5248-8568
Johannes Asplund-Samuelsson http://orcid.org/0000-0001-8077-5305
Elton Paul Hudson http://orcid.org/0000-0003-1899-7649

### Decision letter and Author response

Decision letter https://doi.org/10.7554/eLife.69019.sa1
Author response https://doi.org/10.7554/eLife.69019.sa2

## Additional files

### Supplementary files

• Transparent reporting form

• Supplementary file 1. Table summarizing the constraints for the RBA model.

• Supplementary file 2. Table of all annotated genes for the marked reactions in *Figure 6B–D*. Genes that were found to be essential in one or more conditions are marked with gray background. Fitness values below or above a threshold of $|F| \geq 3$ are marked with red. Non-essential genes annotated for the same reactions were included for comparison.

• Supplementary file 3. Table of the oligonucleotides/primers used in this study.

### Data availability

The mass spectrometry proteomics data have been deposited to the ProteomeXchange Consortium via the PRIDE partner repository with the dataset identifier PXD024819. Protein quantification results can be browsed and interactively analyzed using the web application available at https://m-jahn.shinyapps.io/ShinyProt. Sequencing data for TnSeq and BarSeq experiments are available at the European Nucleotide Archive with accession number PRJEB43757. The data for competition experiments performed with the transposon mutant library can be browsed and interactively analyzed using the web application available at https://m-jahn.shinyapps.io/ShinyLib/.

The following dataset was generated:

| Author(s) | Year | Dataset title | Dataset URL | Database and Identifier |
|---|---|---|---|---|
| Jahn M, Crang N, Janasch M, Hober A, Forsström B, Kimler K, Mattausch A, Chen Q, Asplund-Samuelsson J, Hudson EP | 2021 | Protein allocation and utilization in the versatile chemolithoautotroph Cupriavidus necator | https://www.ebi.ac.uk/pride/archive?keyword=PXD024819 | PRIDE, PXD024819 |
| Jahn M, Crang N, Janasch M, Hober A, Forsström B, Kimler K, Mattausch A, Chen Q, Asplund-Samuelsson J, Hudson EP | 2021 | Gene fitness in the versatile chemolithoautotroph Cupriavidus necator | https://www.ebi.ac.uk/ena/browser/view/PRJEB43757 | European Nucleotide Archive, PRJEB43757 |

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
