## [Editor Report]

This work combines elegant experimental approaches with modelling predictions to
study metabolic adaptations in the bacterium *Cupriavidus necator*, a
microorganism of interest given its metabolic versatility and potential industrial
applications. This manuscript will be interesting for microbiologists and systems
biologists who want to understand how protein production and economy and enzyme
utilization differs in a versatile microorganism in different conditions

---

## [Decision Letter]

**Decision letter after peer review:**

Thank you for submitting your article "Protein allocation and utilization in the
versatile chemolithoautotroph *Cupriavidus necator*" for
consideration by *eLife*. Your article has been reviewed by 3 peer
reviewers, and the evaluation has been overseen by Gisela Storz serving as Reviewing
and Senior Editor. The following individuals involved in review of your submission
have agreed to reveal their identity: Jörg Toepel (Reviewer #1); Nico Claassens
(Reviewer #2).

Essential revisions:

– Determine biomass per OD600 for different conditions.

– Examine PHB production, particularly if the authors want to include the ammonia
condition. Otherwise maybe they should leave out claims regarding growth on that
condition.

– As detailed in the individual reviews, the text would be improved by some
rewriting, including some tempering of conclusions.

*Reviewer #1 (Recommendations for the authors):*

Introduction:

Try to replace wondering at least some times, it sounds very repetitive.

Results:

Figure 3: The significant tests are just reveal that the differences are just between
two groups and not between all of the groups, please indicate that, or did I miss
something?

Discussion:

The conclusion regarding the up-regulation of the cbb cycle under heterotrophic
growth is weak and not supported by the data and controversy to the cited paper. The
authors should state that in detail (as they did but it i am not sure if
sufficient).

What are the consequences for biotechnological applications? Deletion of large parts
of the genome? Please explain and give some ideas.

Delete the last sentence of the discussion it is somehow arbitrary. You also can
explain it in a separate paragraph in more detail

Methods:

Biomass per OD highly depends on the growth condition and bears the risk or error

*Reviewer #2 (Recommendations for the authors):*

The paper is well-written , and could be interesting for the readership of
*eLife* if several major issues are tackled by the authors
first.

1. One of the only clear ' questions' the researchers are after is to determine if
co-utilization of the CCB cycle for growth on fructose is useful. The RBA and
fitness analysis shows this is not the case for fructose and any of the other (even
more oxidized) substrates. This could also be already expected theoretically. If you
consider fructose has a reduction degree per carbon of 4 and biomass about 4.2, it
is clear that no 'electrons' are left over to reduce extra CO2. This would only be
interesting on very reduced substrates like glycerol (degree of reduction >
biomass). However, real biology may behave differently than what 'theoretical
stoichiometries predict', *C. necator* may perform' overoxidation in
the TCA cycle, and this could potentially explain a benefit of the CO2 refixation,
even though on paper it's a wasteful detour. Other papers suggest this overactivity
of TCA cycle, also in autotrophic growth of *C. necator* on hydrogen
(10.1016/j.procbio.2017.07.007), where limiting the oxygen concentration increased
biomass yields. A similar situation may be going on in heterotrophic growth and
explain observations on a small impact. I suggest the authors look at that more
closely and revisit some conclusions on this.

2. The authors state that the deletion of ccbR has no fitness cost for growth on
fructose, however in Figure 5F I still observe a minor, but significant fitness cost
for growth on fructose, hence I suggest to rephrase the text slightly to better
reflect that. This may actually change the conclusion of the paper on the 'useless'
role of the CCB on fructose and sucrose.

3. The authors should better explain the differences between their findings and
Shimizu et al., 2015 paper in Scientific reports.

4. Page 16, line 19 and 32: please check if you are not referring to figure 4
instead.

5. P21 L17 " Another conditionally essential gene on formate was ppc,

encoding the PEP-carboxylase (PPC). The reaction has no other annotated (iso-)
enzymes and was predicted by RBA to carry most flux on formate compared to the other
carbon sources (Figure 6 B)." It would be interesting to explain the
biochemical logic why this anaplerotic reaction would be used more than in other
conditions, where anaplerosis is also needed. I guess it has to do with the fact
that there is generally lower TCA flux during growth on formate, NADH is generated
directly from NADH rather than the TCA cycle. However, anaplerosis is needed in all
conditions to generate oxaloacetate etc. So interesting to explain this.

6. P25 L14 " P25 L14 " Of all central carbon metabolism, the TCA cycle
enzymes

showed on average lowest abundance, variability and utilization. This is similar to
*E. coli*, where the sole flux capacity demand suggested lower
enzyme abundance than what was measured experimentally [Noor et al., 2016]. Only
when reverse flux (reactions with low thermodynamic driving force) and l ow enzyme
saturation, estimated from metabolite levels was taken in to account, was the
calculated enzyme demand similar to the measured levels [Noor et al.,
2016]."

This is an important point the authors raise in the discussion, and also makes wonder
why the authors did not take into account reverse fluxes in their RBA analysis, is
this because the lack of kinetic parameters known, thermodynamic parameters to
perform this also for *C. necator* should be available generally,
right? Good to specify this (or repeat that analysis including reverse fluxes, e.g.
only based on thermodynamics).

7. Be clearer about what sugars *C. necator* can use. On P25 L25 you
state " For *C.*

*necator*, hexose sugars are only one out of many possible substrate
classes and the flux" However, it can only accept one hexose sugar (fructose)
as is stated before twice:

– " The only sugar that supports growth 27 is fructose, which i s metabolized
via the Entner-Doudoroff (ED) pathway [Alagesan et al., 28 2018]."

– " Fructose was chosen as it is the only known sugar (apart from sugar
alcohols) that *C. necator* utilizes [Orita et al., 2012]."

Overall, this is not so clear, Orita et al. do not discuss sugar alcohols and I am
not sure which sugar alcohols *C. necator* can consume (it can
consume some sugar acids, but that's something different, at least glycolate). In
addition, *C. necator* does consume N-acetylglucosamine, which is
defined as sugar. Maybe just rephrase that *C. necator* is only known
to grow on one hexose sugar, being fructose and not on glucose. And a better
reference could be the 2008 review of Cramm, which states " Organic carbon and
energy sources for heterotrophic growth include TCA cycle intermediates, sugar
acids, fatty acids, amino acids, alcohols, and aromatic compounds, while utilization
of sugars is restricted to fructose and N -acetylglucosamine [Johnson and Stanier,
1971; Kersters and De Ley, 1984]"

8. The authors speculate about the evolutionary nature of the two copies of the
Calvin cycle operon, as well as their data show the key role of cbbR. Can the
authors maybe also mention and discuss the presence of an inactive ccbR copy in the
2nd CCB operon on pHG1, which is now not mentioned, but also indicates the 'recent'
and ' unoptimized' acquisition of these CBB operons. Would also be good to know what
is the abundance of this inactive cbbR copy from pHG1 in the proteome?

*Reviewer #3 (Recommendations for the authors):*

This study dealt with comprehensive analyses regarding protein quantitation and
enzyme utilization in the versatile chemolithoautotrophic bacterium *C.
necator*. The experiments and analyses were well considered and
designed, and the authors obtained quite interesting information such as presence of
many under-utilized enzymes as well as excess amount of utilized enzymes, suggesting
the highly robust properties of this bacterium against environmental
perturbations.

I have no comments in the experiments, results, and discussion obtained by
carbon-limited chemostat cultivation, although I here address a few concerns in the
nitrogen-limited cultivation as below, which should be considered by the authors
before publication.

1) It has been well known that *C. necator* H16 synthesizes and
accumulates PHB with in the cells under nitrogen-limited conditions. In many cases
of PHB-producing cells, the accumulation of intracellular PHB granule reflects on
apparent OD owing to the changes in cell size and morphology. In the
nitrogen-limited cultivation in this study, I think that the constant OD was a
result of steady states of both cell growth and PHB synthesis. Did the authors
determine PHB accumulation under the nitrogen-limited condition?

2) PHB is a water-insoluble polyester, thus it can be escaped from cytosolic
equilibrium once after polymerized from water-soluble monomers. Did the RBA model
used in this study include PHB synthesis? That is, in the nitrogen-limited
condition, did the biomass yield calculated by RBA simulation contain PHB?

3) The authors concluded that reassimilation of CO2 by CBB pathway does not provide a
fitness benefit for heterotrophic growth. I agreed with this conclusion, because it
is feasible that energy- and reducing equivalents-consuming CBB cycle is not
required when carbons other than CO2 are available. While, I read the previous
report by Shimizu et al. They demonstrated advantage of CBB cycle under
heterotrophic growth in synthesis of the storage polyester unassociated with cell
growth, and never proposed advantage in the heterotrophic growth. It is not adequate
to discuss about this matter by applying the results in the current study focusing
on growth to the previous study focusing on growth-unassociated PHB synthesis. I
recommend the authors to correct/modify the relevant descriptions in order to avoid
readers' misunderstanding.

[Editors' note: further revisions were suggested prior to acceptance, as described
below.]

Thank you for resubmitting your work entitled "Protein allocation and
utilization in the versatile chemolithoautotroph *Cupriavidus
necator*" for further consideration by *eLife*. Your
revised article has been reviewed by 2 peer reviewers and the evaluation has been
overseen by Gisela Storz serving as Senior and Reviewing Editor.

I very much appreciate the effort you have taken to carefully address the previous
reviews. There are just a few more points outlined below that should be considered
in the final version for *eLife*.

*Reviewer #2 (Recommendations for the authors):*

The authors did a very good job in revising the manuscript and adding additional
experiments and modelling on dry weight and PHB production.

The only answer that puzzled us was the answer on PEP carboxylase, we were puzzled by
the explanation that oxaloacetate flux goes to glycine via reserve GCV operation
(does that mean that this goes via threonine).

Furthermore, they satisfied all our queries well, so I recommend publication.

*Reviewer #3 (Recommendations for the authors):*

I respect the authors' great effort in re-examination of bioreactor experiment and
replacement of metabolic modeling regarding the N-limitation condition. I agree with
the most revisions so think that the revised manuscript is valuable for publication
by *eLife*, but I have still a few concerns as below, which are
expected to be considered by the authors prior to the publication.

1) Were the g-DCW values used for calculation of m-protein (g/g-DCW) in Figure 4, Y
(gDCW/g-S) in Figure 5C (and so on) under the N-limitation condition the cell mass
excluding PHB (PHB-subtracted biomass)? The m-protein graph under the N-limitation
in Figure 4 in the revised manuscript looks like to be the same as the previous
version.

2) The rate of intracellular PHB synthesis usually becomes maximum when the cell
growth has been stopped by nitrogen depletion, so the increase in PHB production by
the function of CBB cycle during the heterotrophic condition on fructose, reported
by Shimizu et al., may become significant during the growth-unassociated PHB
synthesis. The authors carried re-simulation to estimate the advantage of the
heterotrophic CBB cycle (lines 373-385 in the revised manuscript), but this
re-simulation was done still under growth conditions. Is it possible to simulate the
growth-unassociated PHB synthesis on fructose under N-depletion? Fructose-uptake by
*C. necator* was supposed to be weakened after the growth phase
due to marked down-regulation of expression of EM and ED pathway genes (reported in
previous transcriptome analyses). Considering this, I'm not sure whether the
constant fructose uptake rate of 4.0 mmol/gDCW/h was adequate or not for the
growth-unassociated PHB synthesis.

---

## [Author Response]

Essential revisions:– Determine biomass per OD600 for different conditions.

The biomass in g dry cell weight (DCW) was determined for all substrates. For
fructose, succinate and formate, biomass yield was determined using shake flask
experiments, as the biomass yield was constant in previous chemostat cultivations
(no change in optical density at different growth rates). For nitrogen limitation,
such a change in optical density was observed, indicating an increase in total
biomass with increasing N-limitation rate due to the production of PHB. The
ammonium-limited cultivations were therefore reproduced in chemostats and DCW as
well as PHB concentration determined for each growth rate individually.

– Examine PHB production, particularly if the authors want to include the ammonia
condition. Otherwise maybe they should leave out claims regarding growth on that
condition.

The PHB and DCW measurements for all substrates were added to the manuscript as new
Figure 1—figure supplement 2. The new results show that *C. necator*
does not produce PHB during carbon limitation, while it does produce an increasing
amount of PHB with decreasing growth rate in the nitrogen-limited chemostat (up to
80% of biomass was PHB), in line with previous findings in the literature. The
metabolic modeling was updated with the corrected biomass and PHB concentrations. In
summary, the following changes were made to the manuscript:

– yield and substrate uptake rates are now lower than the previous estimation

– some fitted model parameters (e.g. k_app_) have slightly changed due to
changed input

– PHB production was explicitly included in the RBA model

– some figures show slight changes due to updated modeling results

– new Figure 1—figure supplement 2 summarizes PHB and DCW quantification

– new Figure 3—figure supplement 2 shows enzyme utilization of PHB biosynthesis
pathway

– Figure 3D was updated with detailed mass and utilization per substrate instead of
average values

– As detailed in the individual reviews, the text would be improved by some
rewriting, including some tempering of conclusions.

All criticized text sections were changed according to the reviewer's suggestions as
outlined in our response to the reviewer's comments.

Reviewer #1 (Recommendations for the authors):Introduction:Try to replace wondering at least some times, it sounds very repetitive.

The wording was changed in the introduction.

Results:Figure 3: The significant tests are just reveal that the differences are just
between two groups and not between all of the groups, please indicate that, or
did I miss something?

Yes, the significance test is a pairwise comparison of two groups (moderate and high
utilization) to the reference (low utilization). We have added horizontal bars to
Figures 3A and B to indicate which groups were compared, and added an explanation to
the figure legend.

Discussion:The conclusion regarding the up-regulation of the cbb cycle under heterotrophic
growth is weak and not supported by the data and controversy to the cited paper.
The authors should state that in detail (as they did but it i am not sure if
sufficient).

We assume that this criticism refers to our original conclusion that the expression
of the *cbb* operon in heterotrophic conditions (fructose) does not
provide a fitness benefit. This issue was also raised by the other reviewers. We
have toned down our conclusions regarding the benefit of the *cbb*
operon after considering the ambiguity of some of the fitness results, and
revisiting the claims from Shimizu et al. (see more detailed response to comments
from reviewer 2 and 3).

What are the consequences for biotechnological applications? Deletion of large
parts of the genome? Please explain and give some ideas.

We have added the following paragraph about biotechnological implications of our work
to the discussion:

“Our results imply that *C. neactor* is in its current state far from
being an ideal host for biotech applications. […] Alternatively, laboratory
evolution could be employed to select mutants with beneficial traits such as
tolerance to formic acid.”

Delete the last sentence of the discussion it is somehow arbitrary. You also can
explain it in a separate paragraph in more detail

The sentence was removed.

Methods:Biomass per OD highly depends on the growth condition and bears the risk or
error

We agree with the reviewer. The reason we used OD measurement as a proxy for biomass
concentration was the limited volume of our chemostat cultures (65 mL); dry cell
weight measurement makes it necessary to sacrifice a substantial amount of culture
volume. We have now performed a new set of cultivations for all carbon and nitrogen
limitations (fructose, succinate, formate in shake flasks, ammonium in chemostat),
and determined g DCW and PHB from these cultures. The updated substrate uptake rates
and the PHB production were included in the model.

Reviewer #2 (Recommendations for the authors):The paper is well-written , and could be interesting for the readership of eLife
if several major issues are tackled by the authors first.1. One of the only clear ' questions' the researchers are after is to determine
if co-utilization of the CCB cycle for growth on fructose is useful. The RBA and
fitness analysis shows this is not the case for fructose and any of the other
(even more oxidized) substrates. This could also be already expected
theoretically. If you consider fructose has a reduction degree per carbon of 4
and biomass about 4.2, it is clear that no 'electrons' are left over to reduce
extra CO2. This would only be interesting on very reduced substrates like
glycerol (degree of reduction > biomass). However, real biology may behave
differently than what 'theoretical stoichiometries predict', C. necator may
perform' overoxidation in the TCA cycle, and this could potentially explain a
benefit of the CO2 refixation, even though on paper it's a wasteful detour.
Other papers suggest this overactivity of TCA cycle, also in autotrophic growth
of C. necator on hydrogen (10.1016/j.procbio.2017.07.007), where limiting the
oxygen concentration increased biomass yields. A similar situation may be going
on in heterotrophic growth and explain observations on a small impact. I suggest
the authors look at that more closely and revisit some conclusions on this.

We agree with the reviewer that theoretical considerations like degree of reduction
and reaction stoichiometry can already give hints on the usefulness of the CBB cycle
on different substrates. We have included a comment on the degree of reduction for
substrate (fructose) and product (biomass). Other factors such as utilization of
more protein-(in-)efficient pathways can under some circumstances improve growth
rate on the cost of biomass yield, or vice versa (Basan et al., Nature, 2015). In
our study we did not find evidence for such a growth rate/biomass yield
tradeoff.

Another factor influencing biomass yield is the activity of futile cycles or other
energy-wasting reactions. The reference from Lu et al., 2017, falls in this
category. Their results show that only O_2_ but no other limitation
resulted in higher energy efficiency (H_2_ consumed per CO_2_)
during lithoautotrophic growth (see also Yu et al., Int J Hyd Energy, 2013,
https://doi.org/10.1016/j.ijhydene.2013.04.153). The final biomass yield was
unchanged between different limitations (~1 g DCW / g H_2_). We also note
that their reported yield and H_2_ energy efficiency was highest in
non-limited growth, so that O_2_ limitation only improved yield in relative
comparison to other limitations. The authors speculated about changes in energy
efficiency being caused by a shift in membrane bound- versus soluble hydrogenase
(affecting NADH/ATP ratio), or increased TCA cycle flux oxidizing biomass precursors
for energy (p. 155, last paragraph). Unfortunately neither of these hypotheses were
tested experimentally.

From our own simulations (Figure 5D), increased flux through the TCA cycle was in
fact one of the predicted consequences of forcing CBB activity during growth on
fructose. *C. necator* seems to have no yield or rate benefit from
that alone. The idea of an overactive TCA cycle is intriguing, where fixed carbon is
reoxidized and NADH used in respiration. However, when we tested the effect of
O_2_ limitation on *C. necator* cultivated in fructose-
and formate-fed turbidostats we did not see increased yield (OD_600 nm_) or
growth rate during oxygen limitation, see Author response image 1. To us, this data would go against the occurrence of
wasteful oxidation of fixed carbon in the TCA cycle during normal conditions.

**Author response image 1. sa2fig1:** O_2_-limitation does not increase biomass yield, but reduces
growth rate. *C. necator* was grown in fructose or formate-fed turbidostats
(0.5 and 1.5 g/L substrate, respectively). The oxygen concentration was
stepwise reduced by mixing air with molecular nitrogen (N_2_).

2. The authors state that the deletion of ccbR has no fitness cost for growth on
fructose, however in Figure 5F I still observe a minor, but significant fitness
cost for growth on fructose, hence I suggest to rephrase the text slightly to
better reflect that. This may actually change the conclusion of the paper on the
'useless' role of the CCB on fructose and sucrose.

This comment was addressed in a previous response.

3. The authors should better explain the differences between their findings and
Shimizu et al., 2015 paper in Scientific reports.

This comment was addressed in a previous response.

4. Page 16, line 19 and 32: please check if you are not referring to figure 4
instead.

The figure references were corrected. We thank the reviewer for the suggestion.

5. P21 L17 " Another conditionally essential gene on formate was ppc,
encoding the PEP-carboxylase (PPC). The reaction has no other annotated (iso-)
enzymes and was predicted by RBA to carry most flux on formate compared to the
other carbonsources (Figure 6 B)." It would be interesting to explain the biochemical
logic why this anaplerotic reaction would be used more than in other conditions,
where anaplerosis is also needed. I guess it has to do with the fact that there
is generally lower TCA flux during growth on formate, NADH is generated directly
from NADH rather than the TCA cycle. However, anaplerosis is needed in all
conditions to generate oxaloacetate etc. So interesting to explain this.

This is an interesting question. In order to answer it, we have closely inspected the
fluxes from PEP to oxaloacetate (OA) *via PPC* and further down to
different amino acid biosynthesis routes. On formate, 75% of flux from PEP went to
OA while only 12% proceeded to pyruvate. Flux through the TCA was generally low on
formate. As the reviewer points out, OA is a hub for amino acid biosynthesis
reactions and around 90% of it was transaminated to aspartate, which was used for
protein biosynthesis (23%), but also further metabolized to other amino acids (thr,
ala, lys, ile, met, asn) and peptidoglycan. However, we also noticed that a large
portion of the flux from OA (40%) went to the serine hydroxymethyltransferase and
the glycine cleavage system (reactions *GHMT3* and
*GLYAMT*), generating glycine and serine by running reverse of
the canonical direction. While this is thermodynamically possible as recently
demonstrated by engineering the reductive glycine cycle in *C.
necator* (Claassens et al., Met Eng, 2020), it is unlikely to happen
without a high concentration of methyl-THF "pushing" these reactions. This
error was corrected by constraining the reaction directionality in the model.

With this change, the flux from PEP to OA on formate is lower (45% from PEP to OA,
55% to pyruvate, see updated Figure 6B). *PPC* nevertheless has a
more important role on formate compared to fructose, which could explain its
stronger effect on fitness. It carries 70% of the total flux towards the TCA cycle
while it only carries 35% on fructose (50% of flux from ED pathway goes to pyruvate
and enters the TCA downstream of *PPC*). In line with this, the
*ppc* knock-out did not only affect growth on formate (fitness =
-4.2, table in Supplementary file 2), but also on fructose (fitness = -2.7), while
there was no effect on succinate (fitness = -0.1). The fitness associated with
*ppc* is therefore directly correlated to the predicted flux
through *PPC*. We have updated the respective text section and Figure
6B-D with the corrected results.

6. P25 L14 " P25 L14 " Of all central carbon metabolism, the TCA cycle
enzymes showed on average lowest abundance, variability and utilization. This is
similar to *E. coli*, where the sole flux capacity demand
suggested lower enzyme abundance than what was measured experimentally [Noor et
al., 2016]. Only when reverse flux (reactions with low thermodynamic driving
force) and l ow enzyme saturation, estimated from metabolite levels was taken in
to account, was the calculated enzyme demand similar to the measured levels
[Noor et al., 2016]."This is an important point the authors raise in the discussion, and also makes
wonder why the authors did not take into account reverse fluxes in their RBA
analysis, is this because the lack of kinetic parameters known, thermodynamic
parameters to perform this also for C. necator should be available generally,
right? Good to specify this (or repeat that analysis including reverse fluxes,
e.g. only based on thermodynamics).

Protein constrained metabolic modeling is a relatively recent development, and only a
handful of frameworks exist. Of the four ones we know (ME-model, RBA, EFTL, GECKO),
only one -EFTL- currently supports thermodynamic constraints. However, the effort to
set up such a model is considerable and the availability of input parameters is
limited. In this work we chose RBA for its good documentation, numerical efficiency
and semi-automated model generation, although it does not include thermodynamic
constraints. Such constraints may be included in future updates.

7. Be clearer about what sugars C. necator can use. On P25 L25 you state "
For C. necator, hexose sugars are only one out of many possible substrate
classes and the flux" However, it can only accept one hexose sugar
(fructose) as is stated before twice:– " The only sugar that supports growth 27 is fructose, which i s
metabolized via the Entner-Doudoroff (ED) pathway [Alagesan et al., 28
2018]."– " Fructose was chosen as it is the only known sugar (apart from sugar
alcohols) that C. necator utilizes [Orita et al., 2012]."Overall, this is not so clear, Orita et al. do not discuss sugar alcohols and I
am not sure which sugar alcohols C. necator can consume (it can consume some
sugar acids, but that's something different, at least glycolate). In addition,
C. necator does consume N-acetylglucosamine, which is defined as sugar. Maybe
just rephrase that C. necator is only known to grow on one hexose sugar, being
fructose and not on glucose. And a better reference could be the 2008 review of
Cramm, which states " Organic carbon and energy sources for heterotrophic
growth include TCA cycle intermediates, sugar acids, fatty acids, amino acids,
alcohols, and aromatic compounds, while utilization of sugars is restricted to
fructose and N -acetylglucosamine [Johnson and Stanier, 1971; Kersters and De
Ley, 1984]"

We agree and have changed the respective paragraphs to make clear that fructose and
N-acetylglucosamine are the only consumed sugars in the wild type. *C.
necator* can also grow on glycerol, which is a sugar alcohol albeit a
short chain one [Alagesan et al., 2018]. The reference to sugar alcohols was
nevertheless removed because it is of no importance for this study.

8. The authors speculate about the evolutionary nature of the two copies of the
Calvin cycle operon, as well as their data show the key role of cbbR. Can the
authors maybe also mention and discuss the presence of an inactive ccbR copy in
the 2nd CCB operon on pHG1, which is now not mentioned, but also indicates the
'recent' and ' unoptimized' acquisition of these CBB operons. Would also be good
to know what is the abundance of this inactive cbbR copy from pHG1 in the
proteome?

We have added a sentence mentioning the inactive, truncated cbbR^P^ copy on
pHG1 to the discussion. The gene of this potential protein is not included in the
standard genome annotation of *C. necator* that we obtained from
Uniprot. We did therefore not detect the gene product in our data. It is possible
though that identical peptides of the cbbR^P^ protein contribute to the
measured abundance of cbbR^C^ if it is expressed.

Reviewer #3 (Recommendations for the authors):This study dealt with comprehensive analyses regarding protein quantitation and
enzyme utilization in the versatile chemolithoautotrophic bacterium C. necator.
The experiments and analyses were well considered and designed, and the authors
obtained quite interesting information such as presence of many under-utilized
enzymes as well as excess amount of utilized enzymes, suggesting the highly
robust properties of this bacterium against environmental perturbations.I have no comments in the experiments, results, and discussion obtained by
carbon-limited chemostat cultivation, although I here address a few concerns in
the nitrogen-limited cultivation as below, which should be considered by the
authors before publication.1) It has been well known that C. necator H16 synthesizes and accumulates PHB
with in the cells under nitrogen-limited conditions. In many cases of
PHB-producing cells, the accumulation of intracellular PHB granule reflects on
apparent OD owing to the changes in cell size and morphology. In the
nitrogen-limited cultivation in this study, I think that the constant OD was a
result of steady states of both cell growth and PHB synthesis. Did the authors
determine PHB accumulation under the nitrogen-limited condition?

There was indeed a subtle change in optical density over the course of the original
cultivation, from higher OD (~0.4) to lower OD (~0.2) when limitation was relieved
(Figure 1—figure supplement 1A). Revisiting this result, it suggests a different PHB
content at different levels of limitation. As commented before, we redid the
nitrogen limitation experiment and determined absolute biomass in g DCW/L and PHB
content in g/g DCW (new Figure 1—figure supplement 2).

2) PHB is a water-insoluble polyester, thus it can be escaped from cytosolic
equilibrium once after polymerized from water-soluble monomers. Did the RBA
model used in this study include PHB synthesis? That is, in the nitrogen-limited
condition, did the biomass yield calculated by RBA simulation contain PHB?

Yes, the RBA model contains the pathway to synthesize PHB. For simplification, the
chain length of the PHB polymer is not taken into account, so that the molar PHB
concentration is identical to the molar concentration of the 3-hydroxybutyrate-CoA
monomer. PHB is then secreted *via* an exchange reaction and does not
count as biomass. In our previous version of the RBA model, we had included a
constant growth-rate dependent flux towards PHB in order to match the simulated
biomass yield with experimentally determined biomass yield. In our revised
manuscript, this inaccurate behavior was replaced with a more realistic simulation
of PHB production based on the new measurements of biomass and PHB concentration.
The flux towards PHB during nitrogen limitation is now explicitly included in the
model.

3) The authors concluded that reassimilation of CO2 by CBB pathway does not
provide a fitness benefit for heterotrophic growth. I agreed with this
conclusion, because it is feasible that energy – and reducing
equivalents-consuming CBB cycle is not required when carbons other than CO2 are
available. While, I read the previous report by Shimizu et al. They demonstrated
advantage of CBB cycle under heterotrophic growth in synthesis of the storage
polyester unassociated with cell growth, and never proposed advantage in the
heterotrophic growth. It is not adequate to discuss about this matter by
applying the results in the current study focusing on growth to the previous
study focusing on growth-unassociated PHB synthesis. I recommend the authors to
correct/modify the relevant descriptions in order to avoid readers'
misunderstanding.

We agree with the reviewer. Our previous discussion of the findings from Shimizu et
al. could make the impression that the authors claimed a general yield or fitness
benefit of the CBB cycle on fructose. The picture is more nuanced in that the study
from Shimizu et al. found an increase in PHB yield (+20.6% compared to a Rubisco
knockout strain), without commenting on total biomass yield. The authors showed that
during growth on sugar, the emitted and reassimilated CO_2_ was
preferentially incorporated into PHB, and likely caused an increase in
PHB-synthesis. Here, we find that fixation of emitted CO_2_ does not
improve biomass yield on sugar.

Our results suggest that any additional activity of the CBB cycle during metabolism
of fructose is a wasteful process, regardless if the carbon is invested into biomass
or PHB. It is possible that the increased PHB yield that the authors reported when
Rubisco is present is achieved on the cost of the (PHB-subtracted) biomass yield.
However, our results did not allow us to draw conclusions on the effect of
heterotrophic CBB activity on PHB or biomass yield. We deem the resolution of the
transposon library experiments too low to decide if loss of CBB activity
(*cbbR* knock-out) confers a fitness advantage or not. We have
therefore toned down our claims regarding the fitness benefit of CBB cycle and
corrected our interpretation of the results from Shimizu et al. as suggested by the
reviewers (see Results section 5, and discussion).

[Editors' note: further revisions were suggested prior to acceptance, as described
below.]

I very much appreciate the effort you have taken to carefully address the
previous reviews. There are just a few more points outlined below that should be
considered in the final version for eLife.Reviewer #2 (Recommendations for the authors):The authors did a very good job in revising the manuscript and adding additional
experiments and modelling on dry weight and PHB production.The only answer that puzzled us was the answer on PEP carboxylase, we were
puzzled by the explanation that oxaloacetate flux goes to glycine via reserve
GCV operation (does that mean that this goes via threonine).Furthermore, they satisfied all our queries well, so I recommend publication.

In the current manuscript the flux towards serine and glycine is taking the expected
route, that means the canonical pathway starting from the glycolysis intermediate
3-phosphoglycerate (3PG) to 3-phosphohydroxypyruvate, O-phosphoserine, serine, and
glycine + methyl-THF. The methyl-THF is ligated with CO_2_ by the glycine
cleavage system to form one additional glycine.

Before we corrected this error (in our original submission), flux towards serine was
taking the route from oxaloacetate to aspartate, aspartate semialdehyde, homoserine,
phospho-homoserine, threonine, 2-amino-3-oxobutanoate, and finally glycine +
acetyl-CoA. Serine would then be synthesized from 2 molecules glycine (one of it
converted to methyl-THF via the glycine cleavage system). While probably possible,
we deem this route unlikely to have flux in reality because it is much longer, and
requires several enzymatic steps where the enzyme abundance and thermodynamic
driving force might not be sufficient.

Reviewer #3 (Recommendations for the authors):I respect the authors' great effort in re-examination of bioreactor experiment
and replacement of metabolic modeling regarding the N-limitation condition. I
agree with the most revisions so think that the revised manuscript is valuable
for publication by eLife, but I have still a few concerns as below, which are
expected to be considered by the authors prior to the publication.1) Were the g-DCW values used for calculation of m-protein (g/g-DCW) in Figure 4,
Y (gDCW/g-S) in Figure 5C (and so on) under the N-limitation condition the cell
mass excluding PHB (PHB-subtracted biomass)? The m-protein graph under the
N-limitation in Figure 4 in the revised manuscript looks like to be the same as
the previous version.

The protein mass in g/gDCW was calculated by multiplying the relative protein mass
fraction (%) with the average total protein content of 0.68 g / gDCW (Park et al.,
BMC, Sys Bio, 2011), with DCW being dry cell weight without PHB. The graph in Figure
4 looks similar to the previous version because the protein quantification is
relative and only depends on the estimated protein content per DCW, which did not
change. The experimentally measured gDCW was only used to determine the substrate
uptake rate but does not influence the proteomics results.

2) The rate of intracellular PHB synthesis usually becomes maximum when the cell
growth has been stopped by nitrogen depletion, so the increase in PHB production
by the function of CBB cycle during the heterotrophic condition on fructose,
reported by Shimizu et al., may become significant during the
growth-unassociated PHB synthesis. The authors carried re-simulation to estimate
the advantage of the heterotrophic CBB cycle (lines 373-385 in the revised
manuscript), but this re-simulation was done still under growth conditions. Is
it possible to simulate the growth-unassociated PHB synthesis on fructose under
N-depletion? Fructose-uptake by C. necator was supposed to be weakened after the
growth phase due to marked down-regulation of expression of EM and ED pathway
genes (reported in previous transcriptome analyses). Considering this, I'm not
sure whether the constant fructose uptake rate of 4.0 mmol/gDCW/h was adequate
or not for the growth-unassociated PHB synthesis.

The reviewer wonders if PHB production can be modeled when growth has stopped
completely. This is an interesting question, but it would require a different
modeling framework. The RBA model that we used here assumes steady state growth
(which we achieved with chemostat bioreactors) and currently only supports
maximizing growth as objective. The flux towards PHB was forced by constraining the
respective reactions. A model with the ability to predict flux towards PHB without
forcing it would require a dynamic (time-dependent) setup. The objective would need
to be maximizing final biomass, not growth rate, for simulation of a growth
experiment. Such a model would then incentivize storing additional carbon as PHB
during nitrogen limitation, because this strategy would generate more biomass as
soon as nitrogen becomes available again.

Regarding the constant uptake rate of 4 mmol fructose h^-1^
gDCW^-1^ (Figure 4 A-D), we have also tested different
nitrogen/fructose uptake rates in combination with different CO_2_
refixation rates in the RBA model but the results were similar, see Author response image 2. Here, we simulated
a stepwise reduction of the nitrogen concentration which reduces growth and triggers
PHB formation. The model simulations suggest that any additional CBB activity
reduces growth rate and biomass yield even further, and has no advantage for PHB
production. The reason is that – from a purely stoichiometric perspective – the
ratio of CO_2_ re-fixation to fructose uptake rate determines the energetic
efficiency, which decreases with increasing flux through the CBB cycle. In reality,
an increase in PHB yield is nevertheless possible on the cost of the non-PHB biomass
yield.

**Author response image 2. sa2fig2:** Simulation of the combined effect of nitrogen limitation and
CO_2_ refixation via the CBB cycle. Simulations were carried out using the resource balance analysis (RBA) model
for *Ralstonia eutropha* as described previously. Nitrogen
limitation was simulated by stepwise reduction of the nitrogen concentration
in the medium, which triggers (forced) PHB formation. Colors indicate the
amount of flux forced through the Calvin cycle from 0 to 5 mmol
gDCW^-1^ h^-1^. Note that not all simulations were
feasible due to non-positive growth rate. (A) Growth rate. (B) Biomass yield
per g fructose. (C) PHB yield in g per g fructose. (D) Flux in mmol
gDCW^-1^ h^-1^ for selected reactions. FRUabc,
fructose transporter; NH4t, ammonium transporter; EDD, 6PG-hydratase
representing ED flux; CS, citrate synthase representing TCA cycle flux;
RBPC, Rubisco, representing CBB cycle flux; PHBt, PHB pseudo transport
reaction; CO2t, CO_2_ transporter; O2t, O_2_
transporter.